# WHEN DO MODELS GENERALIZE? A PERSPECTIVE FROM DATA-ALGORITHM COMPATIBILITY

## ABSTRACT

One of the major open problems in machine learning is to characterize generalization in the overparameterized regime, where most traditional generalization bounds become inconsistent even for overparameterized linear regression (Nagarajan & Kolter, 2019). In many scenarios, their failure can be attributed to obscuring the crucial interplay between the training algorithm and the underlying data distribution. To address this issue, we propose a concept named compatibility, which quantitatively characterizes generalization in a both data-relevant and algorithm-relevant manner. By considering the entire training trajectory and focusing on early-stopping iterates, compatibility exploits the data and the algorithm information and is therefore a suitable notion for generalization of overparameterized models. We validate this by theoretically studying compatibility under the setting of solving overparameterized linear regression with gradient descent. Specifically, we perform a data-dependent trajectory analysis and derive a sufficient condition for compatibility in such a setting. Our theoretical results demonstrate that in the sense of compatibility, generalization holds with significantly weaker restrictions on the problem instance than the previous at-convergence analysis.

## 1 INTRODUCTION

Although deep neural networks achieve great success in practice (Silver et al., 2017; Devlin et al., 2019; Brown et al., 2020), their remarkable generalization ability is still among the essential mysteries in the deep learning community. One of the most intriguing features of deep neural networks is overparameterization, which confers a level of tractability to the training problem, but leaves traditional generalization theories failing to work. In generalization analysis, both the training algorithm and the data distribution play essential roles (Jiang et al., 2020). For instance, a line of work (Zhang et al., 2021; Nagarajan & Kolter, 2019) highlights the role of the algorithm by showing that the algorithm-irrelevant uniform convergence bounds can become inconsistent in deep learning regimes. Another line of work (Bartlett et al., 2019; Tsigler & Bartlett, 2020) on benign overfitting emphasizes the role of data distribution via profound analysis of specific overparameterized models.

Despite the significant role of data and algorithm in generalization analysis, existing theories usually focus on either the data factor (*e.g.*, uniform convergence (Nagarajan & Kolter, 2019) and last iterate analysis (Bartlett et al., 2019; Tsigler & Bartlett, 2020)) or the algorithm factor (*e.g.*, stability-based bounds (Hardt et al., 2016)). [1] Combining both data and algorithm factor into generalization analysis can help derive tighter generalization bounds and explain the generalization ability of over-parameterized models observed in practice. In this sense, a natural question arises:

*How to incorporate both data factor and algorithm factor into generalization analysis?*

To gain insight into the interplay between data and algorithms, we provide motivating examples of a synthetic overparameterized linear regression task and a classification task on the corrupted MNIST dataset in figure 1. In both scenarios, the final iterate with less algorithmic information,

---

[1] The taxonomy of data-dependent techniques does not mean that they totally ignore all the information of algorithm, but mean that it loses some important algorithm information which makes it vacuous in generalization analysis. Similar arguments hold for algorithm-dependent techniques.

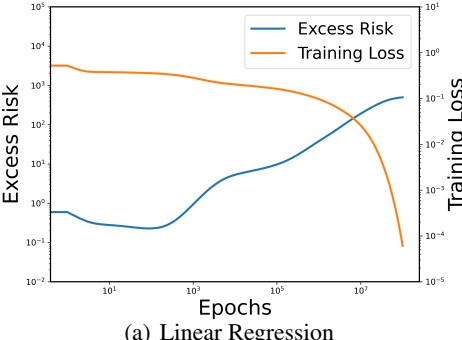
(a) Linear Regression

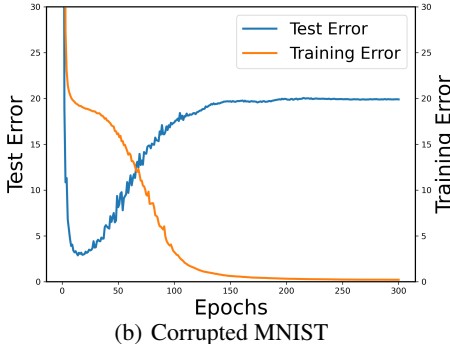
(b) Corrupted MNIST

Figure 1: **(a) The training plot for linear regression with spectrum** $\lambda_i = 1/i^2$ **using GD.** Note that the axes are in the log scale. **(b) The training plot of CNN on corrupted MNIST with 20% label noise using SGD.** Both models successfully learn the useful features in the initial phase of training, but it takes a long time for them to fit the noise in the dataset. The observations demonstrate the power of data-dependent trajectory analysis, since the early stopping solution on the trajectory generalizes well but the final iterate fails to generalize. See Appendix C for details.

which may include the algorithm type (*e.g.*, GD or SGD), hyperparameters (*e.g.*, learning rate, number of epochs), generalizes much worse than the early stopping solutions (see the Blue Line). In the linear regression case, the generalization error of the final iterate can be more than $\times 100$ larger than that of the early stopping solution. In the MNIST case, the final iterate on the SGD trajectory has 19.9% test error, much higher than the 2.88% test error of the best iterate on the GD trajectory. Therefore, the almost ubiquitous strategy of early stopping is a key ingredient in generalization analysis for overparameterized models, whose benefits have been demonstrated both theoretically and empirically (Yao et al., 2007; Ali et al., 2019; Li et al., 2020b; Ji et al., 2021). By focusing on the entire optimization trajectory and performing data-dependent trajectory analysis, both data information and the dynamics of the training algorithm can be exploited to yield consistent generalization bounds.

To analyze the data-dependent trajectory, we introduce a new concept named **data-algorithm-compatibility**, which jointly characterizes the role of the data and the algorithm in generalization analysis. Informally speaking, an algorithm is compatible with a data distribution if as the sample size goes to infinity, the minimum excess risk of the iterates on the training trajectory converges to zero.

The significance of compatibility comes in three folds. Firstly, compatibility incorporates both data and algorithm factors into generalization analysis, and brings new messages into generalization in the overparameterization regime (see Definition 3.1). Secondly, compatibility serves as a minimal condition for generalization, without which one cannot expect to find a consistent solution via standard learning procedures. Consequently, compatibility holds with only mild assumptions and applies to a wide range of problem instances (see Theorem 4.1). Thirdly, compatibility captures the algorithmic significance of early stopping in generalization. By exploiting the algorithm information along the entire trajectory, we arrive at better generalization bounds than the at-convergence analysis (see Table 1 and 2 for examples).

To theoretically validate compatibility, we study it under overparameterized linear regression setting. Analysis of the overparameterized linear regression is a reasonable starting point to study compatibility for more complex models like deep neural networks, since many phenomena of the high dimensional non-linear model are also observed in the linear regime (*e.g.*, Figure 1). Furthermore, the recent neural tangent kernel (NTK) framework demonstrates that very wide neural networks trained using gradient descent with appropriate random initialization can be approximated by kernel regression in a reproducing kernel Hilbert space, which rigorously establishes a close relationship between overparameterized linear regression and deep neural network training (Jacot et al., 2018; Arora et al., 2019).

Specifically, we investigate solving overparameterized linear regression using gradient descent with constant step size, and prove that under some mild regularity conditions, gradient descent is compatible with overparameterized linear regression if the effective dimensions of the feature covariance matrix are asymptotically bounded by the sample size. In this setting, the assumptions needed for generalization in the sense of compatibility are significantly weaker than that in the at-convergence analysis (Bartlett et al., 2019), which demonstrates the benefits of data-relevant and algorithm-relevant generalization analysis.

We summarize our contributions as follows:

- We formalize the notion of compatibility, which highlights the interaction between data and algorithm and serves as a minimal condition for generalization.

- We derive a sufficient condition for compatibility in solving overparameterized linear regression with gradient descent. Our theory substantiates the meaningfulness of compatibility by showing that generalization in the sense of compatibility typically requires much weaker restrictions in the problem instance.

- Technically, we derive time-variant generalization bounds for overparameterized linear regression via data-dependent trajectory analysis. Empirically, various experiment results verify the motivation of compatibility and demonstrate the benefits of early stopping.

## 2 RELATED WORKS

**Data-Dependent Techniques** mainly focus on the data distribution condition for generalization. One of the most popular bounds among data-dependent bounds is uniform convergence (Koltchinskii & Panchenko, 2000; Bartlett et al., 2017; Zhou et al., 2020; Zhang et al., 2021). However, recent works(Nagarajan & Kolter, 2019; Negrea et al., 2020) point out that uniform convergence may not be powerful enough to explain generalization, because it may only yield inconsistent bound in even linear regression cases. Another line of works investigates benign overfitting, which mainly involves generalization at convergence (Bartlett et al., 2019; Zou et al., 2021; Tsigler & Bartlett, 2020; Li et al., 2020c; Wang & Thrampoulidis, 2021; Frei et al., 2022).

**Algorithm-Dependent Techniques** measure the role of the algorithmic information in generalization. A line of works derives generalization bounds via algorithm stability (Hardt et al., 2016; Feldman & Vondrák, 2018; Mou et al., 2018; Feldman & Vondrák, 2019; Bousquet et al., 2020; Li et al., 2020a; Lei & Ying, 2020; Bassily et al., 2020; Teng et al., 2021). A parallel line of works analyzes the implicit bias of algorithmic information (Bousquet & Elisseeff, 2002; Soudry et al., 2018; Shah et al., 2020; Hu et al., 2020; Lyu & Li, 2020; Lyu et al., 2021), which are mainly based on analyzing a specific data distribution (*e.g.*, linear separable).

**Other Generalization Techniques.** Besides the techniques discussed above, there are many other approaches. For example, PAC-Bayes theory performs well empirically and theoretically (Shawe-Taylor & Williamson, 1997; Seeger, 2002; McAllester, 2003; Parrado-Hernández et al., 2012; McAllester, 2013; Dziugaite & Roy, 2017; Neyshabur et al., 2018) and can even yield non-vacuous bounds in deep learning regimes (Rivasplata et al., 2020; Pérez-Ortiz et al., 2021). Furthermore, there are other promising techniques including information theory (Russo & Zou, 2016; Xu & Raginsky, 2017; Banerjee & Montúfar, 2021), and compression-based bounds (Arora et al., 2018).

**Early Stopping** has the potential to improve generalization for various machine learning problems (Raskutti et al., 2014; Vaskevicius et al., 2020; Zhang et al., 2021; Li et al., 2021; Kuzborskij & Szepesvári, 2021; Bai et al., 2021; Shen et al., 2022). A line of works studies the rate of early stopping in linear regression and kernel regression with different algorithms, *e.g.*, gradient descent (Yao et al., 2007), stochastic gradient descent (Tarres & Yao, 2014; Rosasco & Villa, 2015; Dieuleveut & Bach, 2016; Lin & Rosasco, 2017; Pillaud-Vivien et al., 2018), gradient flow (Ali et al., 2019), conjugate gradient (Blanchard & Krämer, 2016) and spectral algorithms (Gerfo et al., 2008; Lin & Cevher, 2018). Of the most relevance here is Yao et al. (2007), which proves an optimal excess bound for a certain class of kernel regression problems solved using early stopping gradient descent, Beyond linear models, early-stopping is also effective for training deep neural networks (Li et al., 2020b; Ji et al., 2021). Another line of research focuses on the signal for early stopping (Prechelt, 2012; Forouzesh & Thiran, 2021).

## 3 COMPATIBILITY

In this section, we formally define compatibility between the data distribution and the training algorithm, starting from the basic notations.

### 3.1 NOTATIONS

**Data Distribution.** Let $\mathcal{D}$ denote the population distribution and $z \sim \mathcal{D}$ denote a data point sampled from distribution $\mathcal{D}$. Usually, $z$ contains a feature and its corresponding response. Besides, we denote the dataset with $n$ samples as $Z \triangleq \{z_i\}_{i \in [n]}$, where $z_i \sim \mathcal{D}$ are i.i.d. sampled from distribution $\mathcal{D}$.

**Loss and Excess Risk.** Let $\ell(\theta; z)$ denote the loss on sample $z$ with parameter $\theta \in \mathbb{R}^p$. The corresponding population loss is defined as $L(\theta; \mathcal{D}) \triangleq \mathbb{E}_{z \sim \mathcal{D}} \ell(\theta; z)$. When the context is clear, we omit the dependency on $\mathcal{D}$ and denote the population loss by $L(\theta)$. Our goal is to find the optimal parameter $\theta^*$ which minimizes the population loss, i.e., $L(\theta^*) = \min_\theta L(\theta)$. Measuring how a parameter $\theta$ approaches $\theta^*$ relies on a term *excess risk* $R(\theta)$, defined as $R(\theta) \triangleq L(\theta) - L(\theta^*)$.

**Algorithm.** Let $\mathcal{A}(\cdot)$ denote a iterative algorithm that takes training data $Z$ as input and outputs a sequence of parameters $\{\theta_n^{(t)}\}_{t \geq 0}$, where $t$ is the iteration number. The algorithm can be either deterministic or stochastic, *e.g.*, variants of (S)GD.

### 3.2 DEFINITIONS OF COMPATIBILITY

Based on the above notations, we introduce the notion of compatibility between data distribution and algorithm in Definition 3.1. Informally, compatibility measures whether a consistent excess risk can be reached along the training trajectory. Note that we omit the role of the loss function in the definition, although the algorithm depends on the loss function.

**Definition 3.1** (**Compatibility**). *Given a loss function $\ell(\cdot)$ with corresponding excess risk $R(\cdot)$, a data distribution $\mathcal{D}$ is **compatible** with an algorithm $\mathcal{A}$ if there exists nonempty subsets $T_n$ of $\mathbb{N}$, such that $\sup_{t \in T_n} R(\theta_n^{(t)})$ converges to zero in probability as sample size $n$ tends to infinity, where $\{\theta_n^{(t)}\}_{t \geq 0}$ denotes the output of algorithm $\mathcal{A}$, and the randomness comes from the sampling of training data $Z$ from distribution $\mathcal{D}$ and the execution of algorithm $\mathcal{A}$. That is to say, $(\mathcal{D}, \mathcal{A})$ is compatible if there exists nonempty sets $T_n$, such that*

$$\sup_{t \in T_n} R(\theta_n^{(t)}) \xrightarrow{P} 0 \quad as \quad n \to \infty. \tag{1}$$

*We call $\{T_n\}_{n>0}$ the compatibility region of $(\mathcal{D}, \mathcal{A})$. The distribution $\mathcal{D}$ is allowed to change with $n$. In this case, $\mathcal{D}$ should be understood as a sequence of distributions $\{\mathcal{D}_n\}_{n \geq 1}$. We also allow the dimension of model parameter $\theta$ to be infinity or to grow with $n$. We omit this dependency on $n$ when the context is clear.*

Compatibility serves as a minimal condition for generalization, since if a data distribution is incompatible with the algorithm, one cannot expect to reach a small excess risk even if we allow for *arbitrary* early stopping. However, we remark that considering only the minimal excess risk is insufficient for a practical purpose, as one cannot exactly find the $t$ that minimizes $R(\theta_n^{(t)})$ due to the noise in the validation set. Therefore, it is meaningful to consider a region of time $t$ on which the excess risk is consistent as in Definition 3.1. The larger the compatibility region is, the more robust the algorithm will be to the noise in its execution.

**Comparisons with Other Notions.** Compared to classic definitions of learnability, *e.g.*, PAC learning, the definition of compatibility is data-specific and algorithm-specific, and is thus a more fine-grained notion. Compared to the concept of *benign* proposed in the recent paper (Bartlett et al., 2019), which studies whether the excess risk at $t = \infty$ converges to zero in probability as the sample size goes to infinity, compatibility only requires that there exists a time to derive a consistent excess risk. We will show later in Section 4.2 that in the overpamameterized linear regression setting, there exist cases such that the problem instance is compatible but not benign.

# 4 COMPATIBILITY ANALYSIS OF OVERPARAMETERIZED LINEAR REGRESSION WITH GRADIENT DESCENT

This paper mainly analyzes compatibility in the overparameterized linear regression regime. We first introduce the data distribution, loss, and training algorithm, and then present the main theorem, which provides a sufficient condition for compatibility in this setting.

## 4.1 PRELIMINARIES FOR OVERPARAMETERIZED LINEAR REGRESSION

**Notations.** Let $O, o, \Omega, \omega$ denote asymptotic notations, with their usual meaning. For example, the argument $a_n = O(b_n)$ means that there exists a large enough constant $C$, such that $a_n \leq Cb_n$. We use $\lesssim$ with the same meaning as the asymptotic notation $O$. Besides, let $\|\boldsymbol{x}\|$ denote the $\ell_2$ norm for vector $\boldsymbol{x}$, and $\|\boldsymbol{A}\|$ denote the operator norm for matrix $\boldsymbol{A}$. We allow the vector to belong to a countably infinite-dimensional Hilbert space $\mathcal{H}$, and with a slight abuse of notation, we use $\mathbb{R}^\infty$ interchangeably with $\mathcal{H}$. In this case, $x^\top z$ denotes inner product and $xz^\top$ denotes tensor product for $x, z \in \mathcal{H}$.

**Data Distribution.** Let $(\boldsymbol{x}, y) \in \mathbb{R}^p \times \mathbb{R}$ denote the feature vector and the response, following a joint distribution $\mathcal{D}$. Let $\boldsymbol{\Sigma} \triangleq \mathbb{E}[\boldsymbol{x}\boldsymbol{x}^\top]$ denote the feature covariance matrix, whose eigenvalue decomposition is $\boldsymbol{\Sigma} = \boldsymbol{V}\boldsymbol{\Lambda}\boldsymbol{V}^\top = \sum_{i>0} \lambda_i \boldsymbol{v}_i \boldsymbol{v}_i^\top$ with decreasing eigenvalues $\lambda_1 \geq \lambda_2 \geq \cdots$. We make the following assumptions on the distribution of the feature vector.

**Assumption 1** (Assumptions on feature distribution). *We assume that*

*1.* $\mathbb{E}[\boldsymbol{x}] = 0$.

*2.* $\lambda_1 > 0, \sum_{i>0} \lambda_i < C$ *for some absolute constant* $C$.

*3. Let* $\tilde{\boldsymbol{x}} = \boldsymbol{\Lambda}^{-\frac{1}{2}} \boldsymbol{V}^\top \boldsymbol{x}$. *The random vector* $\tilde{\boldsymbol{x}}$ *has independent* $\sigma_x$-*subgaussian entries.*[2]

**Loss and Excess Risk.** We choose square loss as the loss function $\ell$, i.e. $\ell(\boldsymbol{\theta}, (\boldsymbol{x}, y)) = 1/2(y - \boldsymbol{x}^\top\boldsymbol{\theta})^2$. The corresponding population loss is denoted by $L(\boldsymbol{\theta}) = \mathbb{E}\ell(\boldsymbol{\theta}, (\boldsymbol{x}, y))$ and the optimal parameter is denoted by $\boldsymbol{\theta}^* \triangleq \operatorname{argmin}_{\boldsymbol{\theta} \in \mathbb{R}^p} L(\boldsymbol{\theta})$. We assume that $\|\boldsymbol{\theta}^*\| < C$ for some absolute constant $C$. If there are multiple such minimizers, we choose an arbitrary one and fix it thereafter. We focus on the excess risk of parameter $\boldsymbol{\theta}$, defined as

$$R(\boldsymbol{\theta}) = L(\boldsymbol{\theta}) - L(\boldsymbol{\theta}^*) = \frac{1}{2}\mathbb{E}(y - \boldsymbol{x}^\top\boldsymbol{\theta})^2 - \frac{1}{2}\mathbb{E}(y - \boldsymbol{x}^\top\boldsymbol{\theta}^*)^2 = \frac{1}{2}(\boldsymbol{\theta} - \boldsymbol{\theta}^*)^\top\boldsymbol{\Sigma}(\boldsymbol{\theta} - \boldsymbol{\theta}^*). \quad (2)$$

Let $\varepsilon = y - \boldsymbol{x}^\top\boldsymbol{\theta}^*$ denote the noise in data point $(\boldsymbol{x}, y)$. The following assumptions involve the conditional distribution of the noise.

**Assumption 2** (Assumptions on noise distribution). *We assume that*

*1. The conditional noise* $\varepsilon|\boldsymbol{x}$ *has zero mean.*

*2. The conditional noise* $\varepsilon|\boldsymbol{x}$ *is* $\sigma_y$-*subgaussian.*

Note that both Assumption 1 and Assumption 2 are commonly considered in the related literatures (Bartlett et al., 2019; Tsigler & Bartlett, 2020; Zou et al., 2021).

**Training Set.** Given a training set $\{(\boldsymbol{x}_i, y_i)\}_{1 \leq i \leq n}$ with $n$ pairs independently sampled from the population distribution $\mathcal{D}$, we define $\boldsymbol{X} \triangleq (\boldsymbol{x}_1, \cdots, \boldsymbol{x}_n)^\top \in \mathbb{R}^{n \times p}$ as the feature matrix, $\boldsymbol{Y} \triangleq (y_1, \cdots, y_n)^\top \in \mathbb{R}^n$ as the corresponding noise vector, and $\boldsymbol{\varepsilon} \triangleq \boldsymbol{Y} - \boldsymbol{X}\boldsymbol{\theta}^*$ as the residual vector. Let the singular value decomposition (SVD) of $\boldsymbol{X}$ be $\boldsymbol{X} = \boldsymbol{U}\tilde{\boldsymbol{\Lambda}}^{\frac{1}{2}}\boldsymbol{W}^\top$, with $\tilde{\boldsymbol{\Lambda}} = \operatorname{diag}\{\mu_1 \cdots, \mu_n\} \in \mathbb{R}^{n \times n}, \mu_1 \geq \cdots \geq \mu_n$.

We consider the overparameterized regime where the feature dimension is larger than the sample size, namely, $p > n$. In this regime, we assume that $\operatorname{rank}(\boldsymbol{X}) = n$ almost surely as in Bartlett et al. (2019). This assumption is equivalent to the invertibility of $XX^\top$.

---

[2]A random variable $X$ is $\sigma$-subgaussian if $\mathbb{E}[e^{\lambda X}] \leq e^{\frac{\lambda^2\sigma^2}{2}}$ for any $\lambda$.

**Assumption 3** (Linear independent training set). *For any $n < p$, we assume that the features in the training set $\{x_1, x_2, \cdots, x_n\}$ is linearly independent almost surely.*

**Algorithm.** Given the dataset $(X, Y)$, define the empirical loss function as $\hat{L}(\theta) \triangleq \frac{1}{2n} \|X\theta - Y\|^2$. We choose full-batch gradient descent on the empirical risk with a constant learning rate $\lambda$ as the algorithm $\mathcal{A}$ in the previous template. In this case, the update rule for the optimization trajectory $\{\theta_t\}_{t \geq 0}$ is formulated as

$$\theta_{t+1} = \theta_t - \frac{\lambda}{n} X^\top (X\theta_t - Y). \tag{3}$$

Without loss of generality, we consider zero initialization $\theta_0 = 0$ in this paper. In this case, for a sufficiently small learning rate $\lambda$, $\theta_t$ converges to the *min-norm interpolator* $\hat{\theta} = X^\top (XX^\top)^{-1} Y$ as $t$ goes to infinity, which was well studied previously (Bartlett et al., 2019). This paper takes one step further and discuss the excess risk along the entire training trajectory $\{R(\theta_t)\}_{t \geq 0}$.

**Effective Rank and Effective Dimensions.** We define the effective rank of the feature matrix $\Sigma$ as $r(\Sigma) \triangleq \frac{\sum_{i>0} \lambda_i}{\lambda_1}$. Our results on compatibility depend on two notions of effective dimension of the feature covariance $\Sigma$, defined as

$$k_0 \triangleq \min \left\{ l \geq 0 : \lambda_{l+1} \leq \frac{c_0 \sum_{i>l} \lambda_i}{n} \right\}, \tag{4}$$

$$k_1 \triangleq \min \left\{ l \geq 0 : \lambda_{l+1} \leq \frac{c_1 \sum_{i>0} \lambda_i}{n} \right\}, \tag{5}$$

where $c_0, c_1$ are constants independent of the dimension $p$, sample size $n$, and time $t$ [3]. We omit the dependency of $k_0, k_1$ on $c_0, c_1, n, \Sigma$ when the context is clear.

## 4.2 COMPATIBILITY FOR OVERPARAMETERIZED LINEAR REGRESSION WITH GRADIENT DESCENT

Next, we present the main result of this section, which provides a clean condition for compatibility between gradient descent and overparameterized linear regression.

**Theorem 4.1** (**Compatibility for Overparameterized Linear Regression with Gradient Descent**). *Consider the overparameterized linear regression setting defined in section 4.1. Let Assumption 1, 2 and 3 hold. Assume the learning rate satisfies $\lambda = O\left(\frac{1}{\text{Tr}(\Sigma)}\right)$. Then under the condition that $k_0 = O(n), k_1 = o(n), r(\Sigma) = o(n)$, gradient descent is compatible with overparameterized linear regression in the region $T_n = \left(\omega\left(\frac{1}{\lambda}\right), o\left(\frac{n}{\lambda}\right)\right)$, namely,*

$$\sup_{t \in T_n} R(\theta_t) \xrightarrow{P} 0 \quad as \quad n \to \infty. \tag{6}$$

*Furthermore, if the feature dimension $p = \infty$ and the data distribution does not change with $n$, then the condition $k_0 = O(n)$ alone suffices for compatibility.*

The proof of Theorem 4.1 is given in Appendix A and sketched in Section 5. The theorem shows that gradient descent is compatible with overparameterized linear regression under some mild regularity conditions on the learning rate, effective rank and effective dimensions. The condition on the learning rate is natural for optimizing a smooth objective. We conjecture that the condition $k_0 = O(n)$ can not be removed in general cases, since the effective dimension $k_0$ characterizes the concentration of the singular values of the data matrix $X$ and plays a crucial role in the excess risk of the gradient descent dynamics. We discuss extensions to the kernel regression setting in Appendix B.6.

**Comparison with Benign Overfitting.** The recent paper Bartlett et al. (2019) studies overparameterized linear regression and gives the condition for min-norm interpolator to generalize. They prove that the feature covariance $\Sigma$ is benign if and only if

$$k_0 = o(n), \quad R_{k_0}(\Sigma) \triangleq \frac{(\sum_{i>k_0} \lambda_i)^2}{\sum_{i>k_0} \lambda_i^2} = \omega(n), \quad r(\Sigma) = o(n) \tag{7}$$

---

[3]Constants may depend on $\sigma_x$, and we omit this dependency thereafter for clarity.

As discussed in Section 3.2, benign problem instance also satisfies compatibility, since benign overfitting requires a stronger condition on $k_0$ and an additional assumption on $R_{k_0}(\Sigma)$. The following example shows that this inclusion relationship is strict.

**Example 4.1.** *Under the same assumption as in Theorem 4.1, if the spectrum of $\Sigma$ satisfies*

$$\lambda_k = \frac{1}{k^\alpha}, \tag{8}$$

*for some $\alpha > 1$, we derive that $k_0 = \Theta(n)$. Therefore, this problem instance satisfies compatibility, but does not satisfy benign overfitting.*

Example 4.1 shows the existence of a case where the early stopping solution can generalize but interpolating solution fails to generalize. Therefore, compatibility can characterize generalization for a much wider range of problem instances. [4]

**Comparisons with Existing Methods.** Theorem 4.1 cannot be directly implied by off-the-shelf stability-based generalization bounds (Hardt et al., 2016; Feldman & Vondrák, 2019) or uniform convergence bounds (Koltchinskii & Panchenko, 2000; Bartlett et al., 2017). The main reason is that both methods rely on a *high probability* bound of the parameter norm, which requires a nontrivial data-dependent and algorithm-dependent analysis. Even if this can be done, both methods will provably give looser bounds with smaller compatibility regions than that in Theorem 4.1. See Appendix B.3 for a detailed comparison with stability-based bounds and Appendix B.4 for discussions on uniform convergence bounds. We also provide a comparison with previous analysis of early stopping (Yao et al., 2007; Lin & Rosasco, 2017; Pillaud-Vivien et al., 2018) in Appendix B.5.

## 5 PROOF SKETCH AND TECHNIQUES

### 5.1 A TIME VARIANT BOUND

We further introduce an additional type of effective dimension besides $k_0, k_1$, which is time variant and is utilized to track the optimization dynamics.

**Definition 5.1** (Effective Dimensions). *Given a feature covariance matrix $\Sigma$, define the effective dimension $k_2$ as*

$$k_2 \triangleq \min \left\{ l \geq 0 : \sum_{i>l} \lambda_i + n\lambda_{l+1} \leq c_2 c(t,n) \sum_{i>0} \lambda_i \right\}, \tag{9}$$

*where $c_2$ is a constant independent of the dimension $p$, sample size $n$, and time $t$. The term $c(t,n)$ is a function to be discussed later. When the context is clear, we omit its dependencies on $c_2, c(t,n), n, \Sigma$ and only denote it by $k_2$.*

Based on the effective rank and effective dimensions defined above, we provide a time-variant bound in Theorem 5.1 for overparameterized linear regression, which further leads to the compatibility argument in Theorem 4.1. Compared to the existing bound (Bartlett et al., 2019), Theorem 5.1 focuses on investigating the role of training epoch $t$ in the excess risk, and is of independent interest.

**Theorem 5.1** (Time Variant Bound). *Suppose Assumption 1, 2 and 3 hold. Fix a function $c(t,n)$. Given $\delta \in (0,1)$, assume that $k_0 \leq \frac{n}{c}$, $\log \frac{1}{\delta} \leq \frac{n}{c}$, $0 < \lambda \leq \frac{1}{c \sum_{i>0} \lambda_i}$ for a large enough constant $c$. Then with probability at least $1 - \delta$, we have for any $t \in \mathbb{N}$,*

$$R(\boldsymbol{\theta}_t) \lesssim B(\boldsymbol{\theta}_t) + V(\boldsymbol{\theta}_t), \tag{10}$$

*where*

$$B(\boldsymbol{\theta}_t) = \|\boldsymbol{\theta}^*\|^2 \left( \frac{1}{\lambda t} + \|\Sigma\| \max \left\{ \sqrt{\frac{r(\Sigma)}{n}}, \frac{r(\Sigma)}{n}, \sqrt{\frac{\log(\frac{1}{\delta})}{n}} \right\} \right), \tag{11}$$

$$V(\boldsymbol{\theta}_t) = \sigma_y^2 \log\left(\frac{1}{\delta}\right) \left( \frac{k_1}{n} + \frac{k_2}{c(t,n)n} + c(t,n) \left( \frac{\lambda t}{n} \sum_{i>0} \lambda_i \right)^2 \right). \tag{12}$$

---

[4]It is worth mentioning that Bartlett et al. (2019) and this paper study the generalization behavior of different models. The comparison here and in the following sections aims to validate the meaningfulness of compatibility analysis, instead of beating their results.

Table 1: **Comparisons of excess risk bound with Bartlett et al. (2019) and Zou et al. (2021).** We provide four types of feature covariance with eigenvalues $\lambda_k$, including *Inverse Polynomial* ($\lambda_k = \frac{1}{k^\alpha}$, $\alpha > 1$), *Inverse Log Polynomial* ($\lambda_k = \frac{1}{k \log^\beta (k+1)}$, $\beta > 1$), *Constant* ($\lambda_k = \frac{1}{n^{1+\varepsilon}}$, $1 \leq k \leq n^{1+\varepsilon}$, $\varepsilon > 0$), and *Piecewise Constant* ($\lambda_k = \frac{1}{s}$ if $1 \leq k \leq s$ and $\lambda_k = \frac{1}{d-s}$ if $s + 1 \leq k \leq d$, where $s = n^r, d = n^q, 0 < r \leq 1, q > 1$). In light of these bounds, ours outperforms Bartlett et al. (2019) in all the cases, and outperforms Zou et al. (2021) in Constant / Piecewise Constant cases if $\varepsilon < \frac{1}{2}$ and $q < \min\{2 - r, \frac{3}{2}\}$. We refer to Appendix B for more details.

| DISTRIBUTIONS | OURS | BARTLETT ET AL. (2019) | ZOU ET AL. (2021) |
|---|---|---|---|
| INVERSE POLYNOMIAL | $O\left(n^{-\min\left\{\frac{\alpha-1}{\alpha}, \frac{1}{2}\right\}}\right)$ | $O(1)$ | $O\left(n^{-\frac{\alpha-1}{\alpha}}\right)$ |
| INVERSE LOG POLYNOMIAL | $O\left(\frac{1}{\log^\beta n}\right)$ | $o(1)$ | $O\left(\frac{1}{\log^\beta n}\right)$ |
| CONSTANT | $O\left(n^{-\frac{1}{2}}\right)$ | $O\left(n^{-\min\left\{\varepsilon, \frac{1}{2}\right\}}\right)$ | $O\left(n^{-\min\{\varepsilon, 1\}}\right)$ |
| PIECEWISE CONSTANT | $O\left(n^{-\min\left\{1-r, \frac{1}{2}\right\}}\right)$ | $O\left(n^{-\min\left\{1-r, q-1, \frac{1}{2}\right\}}\right)$ | $O\left(n^{-\min\{1-r, q-1\}}\right)$ |

We provide a high-level intuition behind Theorem 5.1. We decompose $R(\boldsymbol{\theta}_t)$ into the bias term and the variance term. The variance term is then split into the leading part and tailing part based on the sprctrum of the feature covariance $\boldsymbol{\Sigma}$. The eigenvalues in the tailing part will cause the variance term in the excess risk of the min-norm interpolating solution to be $\Omega(1)$ for fast decaying spectrum, as is the case in (Bartlett et al., 2019). However, since the convergence in the tailing eigenspace is slower compared with the leading eigenspace, a proper early stopping strategy will prevent the overfitting in the tailing eigenspace and meanwhile avoid underfitting in the leading eigenspace.

**The $c(t, n)$ Principle.** It is worth emphasizing that our bound holds for arbitrary positive function $c(t, n)$. Therefore, one can fine-tune the generalization bound by choosing a proper $c(t, n)$. In the subsequent sections, we show how to derive consistent risk bounds for different time $t$, based on different choices of $c(t, n)$. We present the case of choosing a constant $c(t, n)$ in the next section. We leave the case of choosing a varying $c(t, n)$ to Appendix B.7.

## 5.2  VARYING $t$, CONSTANT $c(t, n)$

Theorem 5.1 provides an excess risk upper bound uniformly for $t \in \mathbb{N}$. However, it is still non-trivial to derive Theorem 4.1, where the remaining question is to decide the term $c(t, n)$. The following corollary shows the generalization bound when setting $c(t, n)$ to a constant.

**Corollary 5.1** (Constant $c(t, n)$)**.** *Let Assumption 1, 2 and 3 hold. Fix a constant $c(t, n)$. Suppose $k_0 = O(n)$, $k_1 = o(n)$, $r(\boldsymbol{\Sigma}) = o(n)$, $\lambda = O\left(\frac{1}{\sum_{i>0} \lambda_i}\right)$. Then there exists a sequence of positive constants $\{\delta_n\}_{n \geq 0}$ which converge to 0, such that with probability at least $1 - \delta_n$, the excess risk is consistent for $t \in \left(\omega\left(\frac{1}{\lambda}\right), o\left(\frac{n}{\lambda}\right)\right)$, i.e.*

$$R(\boldsymbol{\theta}_t) = o(1). \tag{13}$$

*Furthermore, for any positive constant $\delta$, with probability at least $1 - \delta$, the minimal excess risk on the training trajectory can be bounded as*

$$\min_t R(\boldsymbol{\theta}_t) \lesssim \frac{\max\{\sqrt{r(\boldsymbol{\Sigma})}, 1\}}{\sqrt{n}} + \frac{\max\{k_1, 1\}}{n}. \tag{14}$$

Lemma 5.1 below shows that $k_1 = o(n)$ always holds for fixed distribution. Therefore, combining Corollary 5.1 and Lemma 5.1 completes the proof of Theorem 4.1.

**Lemma 5.1.** *For any fixed (i.e. independent of sample size $n$) feature covariance $\boldsymbol{\Sigma}$ satisfying assumption 1, we have $k_1(n) = o(n)$.*

**Example Distributions** We apply the bound in Corollary 5.1 to several data distributions. These distributions are widely discussed in (Bartlett et al., 2019; Zou et al., 2021). We also derive the

Table 2: **The effective dimension $k_1$, the optimal early stopping excess risk, and the min-norm excess risk for different feature distributions, with sample size** $n = 100$ , $p = 1000$**.** The table shows that early stopping solutions generalize significantly better than min-norm interpolators, and reveals a positive correlation between the effective dimension $k_1$ and excess risk of early stopping solution. We calculate the 95% confidence interval for each excess risk.

| DISTRIBUTIONS | $k_1$ | OPTIMAL EXCESS RISK | MIN-NORM EXCESS RISK |
|---|---|---|---|
| $\lambda_i = \frac{1}{i}$ | $\Theta(n)$ | $2.399 \pm 0.0061$ | $24.071 \pm 0.2447$ |
| $\lambda_i = \frac{1}{i^2}$ | $\Theta\left(n^{\frac{1}{2}}\right)$ | $0.214 \pm 0.0050$ | $43.472 \pm 0.6463$ |
| $\lambda_i = \frac{1}{i^3}$ | $\Theta\left(n^{\frac{1}{3}}\right)$ | $0.077 \pm 0.0005$ | $10.401 \pm 0.2973$ |
| $\lambda_i = \frac{1}{i \log(i+1)}$ | $\Theta\left(\frac{n}{\log n}\right)$ | $0.697 \pm 0.0053$ | $89.922 \pm 0.9591$ |
| $\lambda_i = \frac{1}{i \log^2(i+1)}$ | $\Theta\left(\frac{n}{\log^2 n}\right)$ | $0.298 \pm 0.0054$ | $82.413 \pm 0.9270$ |
| $\lambda_i = \frac{1}{i \log^3(i+1)}$ | $\Theta\left(\frac{n}{\log^3 n}\right)$ | $0.187 \pm 0.0047$ | $38.145 \pm 0.5862$ |

existing excess risk bounds, which focus on the min-norm interpolator (Bartlett et al., 2019) and one-pass SGD iterate (Zou et al., 2021), of these distributions and compare them with our theorem. The results are summarized in Table 1, which shows that the bound in Corollary 5.1 outperforms previous results for a general class of distributions. We defer the details to Appendix B.

## 6    EXPERIMENTS

In this section, we provide numerical studies of overparameterized linear regression problems. We consider overparameterized linear regression instances with input dimension $p = 1000$, sample size $n = 100$. The features are sampled from Gaussian distribution with different covariances. The empirical results **(a.) demonstrate the benefits of trajectory analysis underlying the definition of compatibility,** since the optimal excess risk along the algorithm trajectory is significantly lower than that of the min-norm interpolator **(b.)  validate the statements in Corollary 5.1**, since the optimal excess risk is lower when the eigenvalues of feature covariance decay faster. We refer to Appendix C for detailed setups, additional results and discussions.

**Observation One: Early stopping solution along the training trajectory generalizes significantly better than the min-norm interpolator.** We calculate the excess risk of optimal early stopping solutions and min-norm interpolators from 1000 independent trials and list the results in Table 2. The results illustrate that the early stopping solution on the algorithm trajectory enjoys much better generalization properties. This observation corroborates the meaningfulness of compatibility and the importance of data-dependent training trajectory in generalization analysis.

**Observation Two: The faster covariance spectrum decays, the lower optimal excess risk is.** Table 2 also illustrates a positive correlation between the decaying rate of $\lambda_i$ and the generalization performance of the early stopping solution. This accords with Theorem 5.1, showing that the excess risk is better for a smaller effective dimension $k_1$, where small $k_1$ indicates a faster-decaying eigenvalue $\lambda_i$. We additionally note that such a phenomenon also illustrates the difference between min-norm and early stopping solutions in linear regression, since Bartlett et al. (2019) demonstrate that the min-norm solution is not consistent when the eigenvalues decay too fast. By comparison, early stopping solutions do not suffer from this restriction.

## 7    CONCLUSION

In this paper, we propose the concept of data-algorithm compatibility, and study compatibility for overparameterized linear regression with gradient descent. Our theoretical and empirical results demonstrate that compatibility eases the assumptions and broadens the scope of generalization by fully exploiting the data information and the algorithm information. Despite linear cases in this paper, compatibility can be a much more general concept. Therefore, we believe this paper will motivate more work on data-dependent trajectory analysis.

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

# Supplementary Materials

## A  PROOFS FOR THE MAIN RESULTS

We first sketch the proof in section A.1 and give some preliminary lemmas A.2. The following sections A.3, A.4 and A.5 are devoted to the proof of Theorem 5.1. The proof of Theorem 4.1 is given in A.6.

### A.1  PROOF SKETCH

We start with a standard bias-variance decomposition following Bartlett et al. (2019), which derives that the time-variant excess risk $R(\boldsymbol{\theta}_t)$ can be bounded by a bias term and a variance term. We refer to Appendix A.3 for more details.

For the bias part, we first decompose it into an optimization error and an approximation error. For the optimization error, we use the spectrum analysis to bound it with $O(1/t)$ where $t$ denotes the time. For the approximation error, we bound it with $O(1/\sqrt{n})$ where $n$ denotes the sample size, inspired by Bartlett et al. (2019). We refer to Appendix A.4 for more details.

For the variance part, a key step is to bound the term $(\boldsymbol{I} - \frac{\lambda}{n}\boldsymbol{X}\boldsymbol{X}^\top)^t$, where $\boldsymbol{X}$ is the feature matrix. The difficulty arises from the different scales of the eigenvalues of $\boldsymbol{X}\boldsymbol{X}^\top$, where the largest eigenvalue has order $\Theta(n)$ while the smallest eigenvalue has order $O(1)$, according to Lemma 10 in Bartlett et al. (2019). To overcome this issue, we divide the matrix $\boldsymbol{X}\boldsymbol{X}^\top$ based on whether its eigenvalues is larger than $c(t,n)$, which is a flexible term dependent on time $t$ and sample size $n$. Therefore, we split the variance term based on eigenvalues of covariance matrix $\boldsymbol{\Sigma}$ (leading to the $k_1$-related term) and based on the eigenvalues of $\boldsymbol{X}\boldsymbol{X}^\top$ (leading to the $k_2$-related term). We refer to Appendix A.5 for more details.

### A.2  PRELIMINARIES

The following result comes from Bartlett et al. (2019), which bounds the eigenvalues of $\boldsymbol{X}\boldsymbol{X}^\top$.

**Lemma A.1.** *(Lemma 10 in Bartlett et al. (2019)) For any $\sigma_x$, there exists a constant c, such that for any $0 \le k < n$, with probability at least $1 - e^{-\frac{n}{c}}$,*

$$\mu_{k+1} \le c\left(\sum_{i>k}\lambda_i + \lambda_{k+1}n\right). \tag{15}$$

This implies that as long as the step size $\lambda$ is small than a threshold independent of sample size $n$, gradient descent is stable.

**Corollary A.1.** *There exists a constant c, such that with probability at least $1 - e^{-\frac{n}{c}}$, for any $0 \le \lambda \le \frac{1}{c\sum_{i>0}\lambda_i}$ we have*

$$\boldsymbol{O} \preceq \boldsymbol{I} - \frac{\lambda}{n}\boldsymbol{X}^\top\boldsymbol{X} \preceq \boldsymbol{I}. \tag{16}$$

*Proof.* The right hand side of the inequality is obvious since $\lambda > 0$. For the left hand side, we have to show that the eigenvalues of $\boldsymbol{I} - \frac{\lambda}{n}\boldsymbol{X}^\top\boldsymbol{X}$ is non-negative. since $\boldsymbol{X}^\top\boldsymbol{X}$ and $\boldsymbol{X}\boldsymbol{X}^\top$ have the same non-zero eigenvalues, we know that with probability at least $1 - e^{-\frac{n}{c}}$, the smallest eigenvalue of $\boldsymbol{I} - \frac{\lambda}{n}\boldsymbol{X}^\top\boldsymbol{X}$ can be lower bounded by

$$1 - \frac{\lambda}{n}\mu_1 \ge 1 - c\lambda\left(\frac{\sum_{i>0}\lambda_i}{n} + \lambda_{k+1}\right) \ge 1 - 2c\lambda\sum_{i>0}\lambda_i \ge 0. \tag{17}$$

where the second inequality uses lemma A.1, and the last inequality holds if $\lambda \le \frac{1}{2c\sum_{i>0}\lambda_i}$.  □

### A.3 PROOF FOR THE BIAS-VARIANCE DECOMPOSITION

Let $\boldsymbol{X}^\dagger$ denote the Moore–Penrose pseudoinverse of matrix $\boldsymbol{X}$. The following lemma gives a closed form expression for $\boldsymbol{\theta}_t$.

**Lemma A.2.** *The dynamics of $\{\boldsymbol{\theta}_t\}_{t\geq 0}$ satisfies*

$$\boldsymbol{\theta}_t = \left(\boldsymbol{I} - \frac{\lambda}{n}\boldsymbol{X}^\top\boldsymbol{X}\right)^t (\boldsymbol{\theta}_0 - \boldsymbol{X}^\dagger\boldsymbol{Y}) + \boldsymbol{X}^\dagger\boldsymbol{Y}. \tag{18}$$

*Proof.* We prove the lemma using induction. The equality holds at $t = 0$ as both sides are $\boldsymbol{\theta}_0$. Recall that $\boldsymbol{\theta}_t$ is updated as

$$\boldsymbol{\theta}_{t+1} = \boldsymbol{\theta}_t + \frac{\lambda}{n}\boldsymbol{X}^\top(\boldsymbol{Y} - \boldsymbol{X}\boldsymbol{\theta}_t). \tag{19}$$

Suppose that the dynamic holds up to the $t$-th step. Plug the expression for $\boldsymbol{\theta}_t$ into the above recursion and note that $\boldsymbol{X}^\top\boldsymbol{X}\boldsymbol{X}^\dagger = \boldsymbol{X}^\top$, we get

$$\begin{aligned}
\boldsymbol{\theta}_{t+1} &= \left(\boldsymbol{I} - \frac{\lambda}{n}\boldsymbol{X}^\top\boldsymbol{X}\right)\boldsymbol{\theta}_t + \frac{\lambda}{n}\boldsymbol{X}^\top\boldsymbol{Y} \\
&= \left(\boldsymbol{I} - \frac{\lambda}{n}\boldsymbol{X}^\top\boldsymbol{X}\right)^{t+1}(\boldsymbol{\theta}_0 - \boldsymbol{X}^\dagger\boldsymbol{Y}) + \left(\boldsymbol{I} - \frac{\lambda}{n}\boldsymbol{X}^\top\boldsymbol{X}\right)\boldsymbol{X}^\dagger\boldsymbol{Y} + \frac{\lambda}{n}\boldsymbol{X}^\top\boldsymbol{Y} \\
&= \left(\boldsymbol{I} - \frac{\lambda}{n}\boldsymbol{X}^\top\boldsymbol{X}\right)^{t+1}(\boldsymbol{\theta}_0 - \boldsymbol{X}^\dagger\boldsymbol{Y}) + \boldsymbol{X}^\dagger\boldsymbol{Y}.
\end{aligned} \tag{20}$$

which finishes the proof. □

Next we prove two identities which will be used in further proof.

**Lemma A.3.** *The following two identities hold for any matrix $X$ and non-negative integer $t$:*

$$\boldsymbol{I} - \boldsymbol{X}^\dagger\boldsymbol{X} + \left(\boldsymbol{I} - \frac{\lambda}{n}\boldsymbol{X}^\top\boldsymbol{X}\right)^t\boldsymbol{X}^\dagger\boldsymbol{X} = \left(\boldsymbol{I} - \frac{\lambda}{n}\boldsymbol{X}^\top\boldsymbol{X}\right)^t, \tag{21}$$

$$\left[\boldsymbol{I} - \left(\boldsymbol{I} - \frac{\lambda}{n}\boldsymbol{X}^\top\boldsymbol{X}\right)^t\right]\boldsymbol{X}^\dagger\boldsymbol{X}\boldsymbol{X}^\top = \boldsymbol{X}^\top\left[\boldsymbol{I} - \left(\boldsymbol{I} - \frac{\lambda}{n}\boldsymbol{X}\boldsymbol{X}^\top\right)^t\right]. \tag{22}$$

*Proof.* Note that $\boldsymbol{X}^\top\boldsymbol{X}\boldsymbol{X}^\dagger = \boldsymbol{X}^\top$, we can expand the left hand side of the first identity above using binomial theorem and eliminate the pseudo-inverse $\boldsymbol{X}^\dagger$:

$$\begin{aligned}
&\boldsymbol{I} - \boldsymbol{X}^\dagger\boldsymbol{X} + \left(\boldsymbol{I} - \frac{\lambda}{n}\boldsymbol{X}^\top\boldsymbol{X}\right)^t\boldsymbol{X}^\dagger\boldsymbol{X} \\
&= \boldsymbol{I} - \boldsymbol{X}^\dagger\boldsymbol{X} + \sum_{k=0}^{t}\binom{t}{k}\left(-\frac{\lambda}{n}\boldsymbol{X}^\top\boldsymbol{X}\right)^k\boldsymbol{X}^\dagger\boldsymbol{X} \\
&= \boldsymbol{I} - \boldsymbol{X}^\dagger\boldsymbol{X} + \boldsymbol{X}^\dagger\boldsymbol{X} + \sum_{k=1}^{t}\binom{t}{k}\left(-\frac{\lambda}{n}\right)^k(\boldsymbol{X}^\top\boldsymbol{X})^{k-1}\boldsymbol{X}^\top\boldsymbol{X}\boldsymbol{X}^\dagger\boldsymbol{X} \\
&= \boldsymbol{I} + \sum_{k=1}^{t}\binom{t}{k}\left(-\frac{\lambda}{n}\right)^k(\boldsymbol{X}^\top\boldsymbol{X})^k \\
&= \left(\boldsymbol{I} - \frac{\lambda}{n}\boldsymbol{X}^\top\boldsymbol{X}\right)^t.
\end{aligned} \tag{23}$$

The second identity can be proved in a similar way:

$$
\left[ \boldsymbol{I} - \left( \boldsymbol{I} - \frac{\lambda}{n} \boldsymbol{X}^\top \boldsymbol{X} \right)^t \right] \boldsymbol{X}^\dagger \boldsymbol{X} \boldsymbol{X}^\top
$$

$$
= -\sum_{k=1}^t \binom{t}{k} \left( -\frac{\lambda}{n} \boldsymbol{X}^\top \boldsymbol{X} \right)^k \boldsymbol{X}^\dagger \boldsymbol{X} \boldsymbol{X}^\top
$$

$$
= -\sum_{k=1}^t \binom{t}{k} \left( -\frac{\lambda}{n} \right)^k (\boldsymbol{X}^\top \boldsymbol{X})^{k-1} \boldsymbol{X}^\top \boldsymbol{X} \boldsymbol{X}^\dagger \boldsymbol{X} \boldsymbol{X}^\top
$$

$$
= -\sum_{k=1}^t \binom{t}{k} \left( -\frac{\lambda}{n} \right)^k (\boldsymbol{X}^\top \boldsymbol{X})^{k-1} \boldsymbol{X}^\top \boldsymbol{X} \boldsymbol{X}^\top \tag{24}
$$

$$
= -\sum_{k=1}^t \binom{t}{k} \left( -\frac{\lambda}{n} \right)^k \boldsymbol{X}^\top (\boldsymbol{X} \boldsymbol{X}^\top)^k
$$

$$
= \boldsymbol{X}^\top \left[ \boldsymbol{I} - \left( \boldsymbol{I} - \frac{\lambda}{n} \boldsymbol{X} \boldsymbol{X}^\top \right)^t \right].
$$

$\square$

We are now ready to prove the main result of this section.

**Lemma A.4.** *The excess risk at the $t$-th epoch can be upper bounded as*

$$
R(\boldsymbol{\theta}_t) \leq \boldsymbol{\theta}^{*\top} \boldsymbol{B} \boldsymbol{\theta}^* + \boldsymbol{\varepsilon}^\top \boldsymbol{C} \boldsymbol{\varepsilon}, \tag{25}
$$

*where*

$$
\boldsymbol{B} = \left( \boldsymbol{I} - \frac{\lambda}{n} \boldsymbol{X}^\top \boldsymbol{X} \right)^t \boldsymbol{\Sigma} \left( \boldsymbol{I} - \frac{\lambda}{n} \boldsymbol{X}^\top \boldsymbol{X} \right)^t, \tag{26}
$$

$$
\boldsymbol{C} = (\boldsymbol{X} \boldsymbol{X}^\top)^{-1} \left[ \boldsymbol{I} - \left( \boldsymbol{I} - \frac{\lambda}{n} \boldsymbol{X} \boldsymbol{X}^\top \right)^t \right] \boldsymbol{X} \boldsymbol{\Sigma} \boldsymbol{X}^\top \left[ \boldsymbol{I} - \left( \boldsymbol{I} - \frac{\lambda}{n} \boldsymbol{X} \boldsymbol{X}^\top \right)^t \right] (\boldsymbol{X} \boldsymbol{X}^\top)^{-1}, \tag{27}
$$

*which characterizes bias term and variance term in the excess risk. Furthermore, there exists constant $c$ such that with probability at least $1 - \delta$ over the randomness of $\boldsymbol{\varepsilon}$, we have*

$$
\boldsymbol{\varepsilon}^\top \boldsymbol{C} \boldsymbol{\varepsilon} \leq c \sigma_y^2 \log \frac{1}{\delta} \operatorname{Tr}[\boldsymbol{C}]. \tag{28}
$$

*Proof.* First note that $\boldsymbol{X} \boldsymbol{X}^\top$ is invertible by Assumption 3. Express the excess risk as follows

$$
\begin{aligned}
R(\boldsymbol{\theta}_t) &= \frac{1}{2} \mathbb{E}[(y - \boldsymbol{x}^\top \boldsymbol{\theta}_t)^2 - (y - \boldsymbol{x}^\top \boldsymbol{\theta}^*)^2] \\
&= \frac{1}{2} \mathbb{E}[(y - \boldsymbol{x}^\top \boldsymbol{\theta}^* + \boldsymbol{x}^\top \boldsymbol{\theta}^* - \boldsymbol{x}^\top \boldsymbol{\theta}_t)^2 - (y - \boldsymbol{x}^\top \boldsymbol{\theta}^*)^2] \\
&= \frac{1}{2} \mathbb{E}[(\boldsymbol{x}^\top (\boldsymbol{\theta}_t - \boldsymbol{\theta}^*))^2 + 2(y - \boldsymbol{x}^\top \boldsymbol{\theta}^*)(\boldsymbol{x}^\top \boldsymbol{\theta}^* - \boldsymbol{x}^\top \boldsymbol{\theta}_t)] \\
&= \frac{1}{2} \mathbb{E}[\boldsymbol{x}^\top (\boldsymbol{\theta}_t - \boldsymbol{\theta}^*)]^2.
\end{aligned} \tag{29}
$$

Recall that $\boldsymbol{\theta}_0 = 0$ and $\boldsymbol{Y} = \boldsymbol{X} \boldsymbol{\theta}^* + \boldsymbol{\varepsilon}$ and we can further simplify the formula for $\boldsymbol{\theta}_t$ in lemma A.2:

$$
\begin{aligned}
\boldsymbol{\theta}_t &= \left( \boldsymbol{I} - \frac{\lambda}{n} \boldsymbol{X}^\top \boldsymbol{X} \right)^t (\boldsymbol{\theta}_0 - \boldsymbol{X}^\dagger \boldsymbol{Y}) + \boldsymbol{X}^\dagger \boldsymbol{Y} \\
&= \left[ \boldsymbol{I} - \left( \boldsymbol{I} - \frac{\lambda}{n} \boldsymbol{X}^\top \boldsymbol{X} \right)^t \right] \boldsymbol{X}^\dagger (\boldsymbol{X} \boldsymbol{\theta}^* + \boldsymbol{\varepsilon}).
\end{aligned} \tag{30}
$$

Plug it into the above expression for $R(\boldsymbol{\theta}_t)$, we have

$$
\begin{aligned}
R(\boldsymbol{\theta}_t) &= \frac{1}{2}\mathbb{E}\left[\boldsymbol{x}^\top\left[\boldsymbol{I} - \left(\boldsymbol{I} - \frac{\lambda}{n}\boldsymbol{X}^\top\boldsymbol{X}\right)^t\right]\boldsymbol{X}^\dagger(\boldsymbol{X}\boldsymbol{\theta}^* + \boldsymbol{\varepsilon}) - \boldsymbol{x}^\top\boldsymbol{\theta}^*\right]^2 \\
&= \frac{1}{2}\mathbb{E}\left[\boldsymbol{x}^\top\left(\boldsymbol{X}^\dagger\boldsymbol{X} - \left(\boldsymbol{I} - \frac{\lambda}{n}\boldsymbol{X}^\top\boldsymbol{X}\right)^t\boldsymbol{X}^\dagger\boldsymbol{X} - \boldsymbol{I}\right)\boldsymbol{\theta}^* \right. \\
&\quad \left. + \boldsymbol{x}^\top\left[\boldsymbol{I} - \left(\boldsymbol{I} - \frac{\lambda}{n}\boldsymbol{X}^\top\boldsymbol{X}\right)^t\right]\boldsymbol{X}^\dagger\boldsymbol{\varepsilon}\right]^2.
\end{aligned}
\tag{31}
$$

Applying lemma A.3, we obtain

$$
\begin{aligned}
R(\boldsymbol{\theta}_t) &= \frac{1}{2}\mathbb{E}\left[-\boldsymbol{x}^\top\left(\boldsymbol{I} - \frac{\lambda}{n}\boldsymbol{X}^\top\boldsymbol{X}\right)^t\boldsymbol{\theta}^* + \boldsymbol{x}^\top\boldsymbol{X}^\top\left[\boldsymbol{I} - \left(\boldsymbol{I} - \frac{\lambda}{n}\boldsymbol{X}\boldsymbol{X}^\top\right)^t\right](\boldsymbol{X}\boldsymbol{X}^\top)^{-1}\boldsymbol{\varepsilon}\right]^2 \\
&\leq \mathbb{E}\left[\boldsymbol{x}^\top\left(\boldsymbol{I} - \frac{\lambda}{n}\boldsymbol{X}^\top\boldsymbol{X}\right)^t\boldsymbol{\theta}^*\right]^2 + \mathbb{E}\left[\boldsymbol{x}^\top\boldsymbol{X}^\top\left[\boldsymbol{I} - \left(\boldsymbol{I} - \frac{\lambda}{n}\boldsymbol{X}\boldsymbol{X}^\top\right)^t\right](\boldsymbol{X}\boldsymbol{X}^\top)^{-1}\boldsymbol{\varepsilon}\right]^2 \\
&:= \boldsymbol{\theta}^{*\top}\boldsymbol{B}\boldsymbol{\theta}^* + \boldsymbol{\varepsilon}^\top\boldsymbol{C}\boldsymbol{\varepsilon}.
\end{aligned}
\tag{32}
$$

which proves the first claim in the lemma. The second part of the theorem directly follows from lemma 18 in Bartlett et al. (2019). $\qquad\square$

## A.4 PROOF FOR THE BIAS UPPER BOUND

The next lemma guarantees that the sample covariace matrix $\frac{1}{n}\boldsymbol{X}^\top\boldsymbol{X}$ concentrates well around $\boldsymbol{\Sigma}$.

**Lemma A.5.** *(Lemma 35 in Bartlett et al. (2019)) There exists constant $c$ such that for any $0 < \delta < 1$ with probability as least $1 - \delta$,*

$$
\left\|\boldsymbol{\Sigma} - \frac{1}{n}\boldsymbol{X}^\top\boldsymbol{X}\right\| \leq c\|\boldsymbol{\Sigma}\|\max\left\{\sqrt{\frac{r(\boldsymbol{\Sigma})}{n}}, \frac{r(\boldsymbol{\Sigma})}{n}, \sqrt{\frac{\log(\frac{1}{\delta})}{n}}, \frac{\log(\frac{1}{\delta})}{n}\right\}.
\tag{33}
$$

The following inequality will be useful in our proof to characterize the decaying rate of the bias term with $t$.

**Lemma A.6.** *For any positive semidefinite matrix $\boldsymbol{P}$ which satisfies $\|\boldsymbol{P}\| \leq 1$, we have*

$$
\|\boldsymbol{P}(1 - \boldsymbol{P})^t\| \leq \frac{1}{t}.
\tag{34}
$$

*Proof.* Assume without loss of generality that $\boldsymbol{P}$ is diagonal. Then it suffices to consider seperately each eigenvalue $\sigma$ of $\boldsymbol{P}$, and show that $\sigma(1 - \sigma)^t \leq \frac{1}{t}$.

In fact, by AM-GM inequality we have

$$
\sigma(1 - \sigma)^t \leq \frac{1}{t}\left[\frac{t\sigma + (1 - \sigma)t}{t + 1}\right]^{t+1} \leq \frac{1}{t},
\tag{35}
$$

which completes the proof. $\qquad\square$

Next we prove the main result of this section.

**Lemma A.7.** *There exists constant $c$ such that if $0 \leq \lambda \leq \frac{1}{c\sum_{i>0}\lambda_i}$, then for any $0 < \delta < 1$, with probability at least $1 - \delta$ the following bound on the bias term holds for any $t$*

$$
\boldsymbol{\theta}^{*\top}\boldsymbol{B}\boldsymbol{\theta}^* \leq c\|\boldsymbol{\theta}^*\|^2\left(\frac{1}{\lambda t} + \|\boldsymbol{\Sigma}\|\max\left\{\sqrt{\frac{r(\boldsymbol{\Sigma})}{n}}, \frac{r(\boldsymbol{\Sigma})}{n}, \sqrt{\frac{\log(\frac{1}{\delta})}{n}}, \frac{\log(\frac{1}{\delta})}{n}\right\}\right).
\tag{36}
$$

*Proof.* The bias can be decomposed into the following two terms

$$\boldsymbol{\theta}^{*\top}\boldsymbol{B}\boldsymbol{\theta}^* = \boldsymbol{\theta}^{*\top}\left(\boldsymbol{I} - \frac{\lambda}{n}\boldsymbol{X}^\top\boldsymbol{X}\right)^t \left(\boldsymbol{\Sigma} - \frac{1}{n}\boldsymbol{X}^\top\boldsymbol{X}\right)\left(\boldsymbol{I} - \frac{\lambda}{n}\boldsymbol{X}^\top\boldsymbol{X}\right)^t \boldsymbol{\theta}^*$$
$$+ \boldsymbol{\theta}^{*\top}\left(\frac{1}{n}\boldsymbol{X}^\top\boldsymbol{X}\right)\left(\boldsymbol{I} - \frac{\lambda}{n}\boldsymbol{X}^\top\boldsymbol{X}\right)^{2t}\boldsymbol{\theta}^*. \tag{37}$$

For sufficiently small learning rate $\lambda$ as given by corollary A.1, we know that with high probability

$$\left\|\boldsymbol{I} - \frac{\lambda}{n}\boldsymbol{X}^\top\boldsymbol{X}\right\| \le 1, \tag{38}$$

which together with lemma A.5 gives a high probability bound on the first term:

$$\boldsymbol{\theta}^{*\top}\left(\boldsymbol{I} - \frac{\lambda}{n}\boldsymbol{X}^\top\boldsymbol{X}\right)^t \left(\boldsymbol{\Sigma} - \frac{1}{n}\boldsymbol{X}^\top\boldsymbol{X}\right)\left(\boldsymbol{I} - \frac{\lambda}{n}\boldsymbol{X}^\top\boldsymbol{X}\right)^t \boldsymbol{\theta}^*$$
$$\le c\|\boldsymbol{\Sigma}\| \|\boldsymbol{\theta}^*\|^2 \max\left\{\sqrt{\frac{r(\boldsymbol{\Sigma})}{n}}, \frac{r(\boldsymbol{\Sigma})}{n}, \sqrt{\frac{\log(\frac{1}{\delta})}{n}}, \frac{\log(\frac{1}{\delta})}{n}\right\}. \tag{39}$$

For the second term, invoke lemma A.6 with $\boldsymbol{P} = \frac{\lambda}{n}\boldsymbol{X}^\top\boldsymbol{X}$ and we get

$$\boldsymbol{\theta}^{*\top}\left(\frac{1}{n}\boldsymbol{X}^\top\boldsymbol{X}\right)\left(\boldsymbol{I} - \frac{\lambda}{n}\boldsymbol{X}^\top\boldsymbol{X}\right)^{2t}\boldsymbol{\theta}^* \le \frac{1}{\lambda}\|\boldsymbol{\theta}^*\|^2 \left\|\left(\frac{\lambda}{n}\boldsymbol{X}^\top\boldsymbol{X}\right)\left(\boldsymbol{I} - \frac{\lambda}{n}\boldsymbol{X}^\top\boldsymbol{X}\right)^{2t}\right\|$$
$$\le \frac{1}{2\lambda t}\|\boldsymbol{\theta}^*\|^2. \tag{40}$$

Putting these two bounds together gives the proof for the main theorem.

$\square$

### A.5 Proof for the Variance Upper Bound

Recall that $\boldsymbol{X} = \boldsymbol{U}\tilde{\boldsymbol{\Lambda}}^{\frac{1}{2}}\boldsymbol{W}^\top$ is the singular value decomposition of data matrix $\boldsymbol{X}$, where $\boldsymbol{U} = (\boldsymbol{u}_1, \cdots, \boldsymbol{u}_n)$, $\boldsymbol{W} = (\boldsymbol{w}_1, \cdots, \boldsymbol{w}_n)$, $\tilde{\boldsymbol{\Lambda}} = \text{diag}\{\mu_1, \cdots, \mu_n\}$ with $\mu_1 \ge \mu_2 \ge \cdots \mu_n$.

Recall that

$$k_0 = \min\{l \ge 0 : \lambda_{l+1} \le \frac{c_0 \sum_{i>l} \lambda_i}{n}\},$$
$$k_1 = \min\{l \ge 0 : \lambda_{l+1} \le \frac{c_1 \sum_{i>0} \lambda_i}{n}\}, \tag{41}$$
$$k_2 = \min\{l \ge 0 : \sum_{i>l} \lambda_i + n\lambda_{l+1} \le c_2 c(t,n) \sum_{i>0} \lambda_i\}\},$$

for some constant $c_0, c_1, c_2$ and function $c(t, n)$.

We further define

$$k_3 = \min\{l \ge 0 : \mu_{l+1} \le c_3 c(t,n) \sum_{i>0} \lambda_i\}, \tag{42}$$

for some constant $c_3$.

The next lemma shows that we can appropriately choose constants to ensure that $k_3 \le k_2$ holds with high probability, and in some specific cases we have $k_2 \le k_1$.

**Lemma A.8.** *For any function $c(t, n)$ and constant $c_2$, there exists constants $c, c_3$, such that $k_3 \le k_2$ with probability at least $1 - e^{-\frac{n}{c}}$. Furthermore, if $c(t, n)$ is a positive constant function, for any $c_1$, there exists $c_2$ such that $k_2 \le k_1$.*

*Proof.* According to lemma A.1, there exists a constant $c$, with probability at least $1 - e^{-\frac{n}{c}}$ we have

$$\mu_{k_2+1} \le c(\sum_{i>k_2} \lambda_i + n\lambda_{k_2+1}) \le cc_2 c(t,n) \sum_{i>0} \lambda_i. \tag{43}$$

Therefore, we know that $k_3 \le k_2$ for $c_3 = cc_2$.

By the definition of $k_1$, we have

$$\sum_{i>k_1} \lambda_i + n\lambda_{k_1+1} \le (c_1 + 1) \sum_{i>0} \lambda_i, \tag{44}$$

which implies that $k_2 \le k_1$ for $c_2 = \frac{c_1+1}{c(t,n)}$, if $c(t,n)$ is a positive constant. □

Next we bound $\mathrm{Tr}[C]$, which implies an upper bound on the variance term.

**Theorem A.1.** *There exist constants $c, c_0, c_1, c_2$ such that if $k_0 \le \frac{n}{c}$, then with probability at least $1 - e^{-\frac{n}{c}}$, the trace of the variance matrix $C$ has the following upper bound for any $t$:*

$$\mathrm{Tr}[C] \le c \left( \frac{k_1}{n} + \frac{k_2}{c(t,n)n} + c(t,n) \left( \frac{\lambda t}{n} \sum_{i>0} \lambda_i \right)^2 \right). \tag{45}$$

*Proof.* We divide the eigenvalues of $XX^\top$ into two groups based on whether they are greater than $c_3 c(t,n) \sum_{i>0} \lambda_i$. The first group consists of $\mu_1 \cdots \mu_{k_3}$, and the second group consists of $\mu_{k_3+1} \cdots \mu_n$. For $1 \le j \le k_3$, we have

$$1 - \left( 1 - \frac{\lambda}{n} \mu_j \right)^t \le 1. \tag{46}$$

Therefore we have the following upper bound on $\left[ I - \left( I - \frac{\lambda}{n} XX^\top \right)^t \right]^2$:

$$
\begin{aligned}
&\left[ I - \left( I - \frac{\lambda}{n} XX^\top \right)^t \right]^2 \\
&= U\mathrm{diag} \left\{ \left[ 1 - \left( 1 - \frac{\lambda}{n}\mu_1 \right)^t \right]^2 \cdots \left[ 1 - \left( 1 - \frac{\lambda}{n}\mu_n \right)^t \right]^2 \right\} U^\top \\
&\preceq U\mathrm{diag} \left\{ \overbrace{1, \cdots 1}^{k_3 \text{ times}}, \overbrace{\left[ 1 - \left( 1 - \frac{\lambda}{n}\mu_{k_3+1} \right)^t \right]^2, \cdots \left[ 1 - \left( 1 - \frac{\lambda}{n}\mu_n \right)^t \right]^2}^{n-k_3 \text{ times}} \right\} U^\top \\
&= U\mathrm{diag} \left\{ \overbrace{1, \cdots 1}^{k_3 \text{ times}}, \overbrace{0, \cdots 0}^{n-k_3 \text{ times}} \right\} U^\top \\
&\quad + U\mathrm{diag} \left\{ \overbrace{0, \cdots 0}^{k_3 \text{ times}}, \overbrace{\left[ 1 - \left( 1 - \frac{\lambda}{n}\mu_{k_3+1} \right)^t \right]^2, \cdots \left[ 1 - \left( 1 - \frac{\lambda}{n}\mu_n \right)^t \right]^2}^{n-k_3 \text{ times}} \right\} U^\top.
\end{aligned}
\tag{47}
$$

For positive semidefinite matrices $P, Q, R$ which satisfies $Q \preceq R$, it holds that $\mathrm{Tr}[PQ] \le \mathrm{Tr}[PR]$. It implies the following upperbound of $\mathrm{Tr}[C]$:

$$
\begin{aligned}
&\mathrm{Tr}[\boldsymbol{C}] \\
&= \mathrm{Tr}\left[\left[\boldsymbol{I} - \left(\boldsymbol{I} - \frac{\lambda}{n}\boldsymbol{X}\boldsymbol{X}^\top\right)^t\right]^2 \left(\boldsymbol{X}\boldsymbol{X}^\top\right)^{-2}\boldsymbol{X}\boldsymbol{\Sigma}\boldsymbol{X}^\top\right] \\
&\leq \underbrace{\mathrm{Tr}\left[\boldsymbol{U}\,\mathrm{diag}\left\{\overbrace{1,\cdots 1}^{k_3 \text{ times}},\ \overbrace{0,\cdots 0}^{n-k_3 \text{ times}}\right\}\boldsymbol{U}^\top\left(\boldsymbol{X}\boldsymbol{X}^\top\right)^{-2}\boldsymbol{X}\boldsymbol{\Sigma}\boldsymbol{X}^\top\right]}_{\textcircled{1}} \\
&\quad + \underbrace{\mathrm{Tr}\left[\boldsymbol{U}\,\mathrm{diag}\left\{\overbrace{0,\cdots 0}^{k_3 \text{ times}},\ \overbrace{\left[1-\left(1-\frac{\lambda}{n}\mu_{k_3+1}\right)^t\right]^2,\cdots\left[1-\left(1-\frac{\lambda}{n}\mu_n\right)^t\right]^2}^{n-k_3 \text{ times}}\right\}\boldsymbol{U}^\top\left(\boldsymbol{X}\boldsymbol{X}^\top\right)^{-2}\boldsymbol{X}\boldsymbol{\Sigma}\boldsymbol{X}^\top\right]}_{\textcircled{2}}.
\end{aligned}
\tag{48}
$$

**Bounding ①**

Noticing $\boldsymbol{X} = \boldsymbol{U}\tilde{\boldsymbol{\Lambda}}^{\frac{1}{2}}\boldsymbol{W}^\top$ and $\boldsymbol{\Sigma} = \sum_{i\geq 1}\lambda_i\boldsymbol{v}_i\boldsymbol{v}_i^\top$, we express the first term as sums of eigenvector products,

$$
\begin{aligned}
\textcircled{1} &= \mathrm{Tr}\left[\boldsymbol{U}\,\mathrm{diag}\left\{\overbrace{1,\cdots 1}^{k_3 \text{ times}},\ \overbrace{0,\cdots 0}^{n-k_3 \text{ times}}\right\}\boldsymbol{U}^\top\left(\boldsymbol{X}\boldsymbol{X}^\top\right)^{-2}\boldsymbol{X}\boldsymbol{\Sigma}\boldsymbol{X}^\top\right] \\
&= \mathrm{Tr}\left[\boldsymbol{U}\,\mathrm{diag}\left\{\overbrace{1,\cdots 1}^{k_3 \text{ times}},\ \overbrace{0,\cdots 0}^{n-k_3 \text{ times}}\right\}\boldsymbol{U}^\top\boldsymbol{U}\tilde{\boldsymbol{\Lambda}}^{-2}\boldsymbol{U}^\top\boldsymbol{U}\tilde{\boldsymbol{\Lambda}}^{\frac{1}{2}}\boldsymbol{W}^\top\boldsymbol{\Sigma}\boldsymbol{W}\tilde{\boldsymbol{\Lambda}}^{\frac{1}{2}}\boldsymbol{U}^\top\right] \\
&= \mathrm{Tr}\left[\mathrm{diag}\left\{\overbrace{1,\cdots 1}^{k_3 \text{ times}},\ \overbrace{0,\cdots 0}^{n-k_3 \text{ times}}\right\}\tilde{\boldsymbol{\Lambda}}^{-1}\boldsymbol{W}^\top\boldsymbol{\Sigma}\boldsymbol{W}\right] \\
&= \sum_{i\geq 1}\lambda_i\,\mathrm{Tr}\left[\mathrm{diag}\left\{\overbrace{1,\cdots 1}^{k_3 \text{ times}},\ \overbrace{0,\cdots 0}^{n-k_3 \text{ times}}\right\}\tilde{\boldsymbol{\Lambda}}^{-1}\boldsymbol{W}^\top\boldsymbol{v}_i\boldsymbol{v}_i^\top\boldsymbol{W}\right] \\
&= \sum_{i\geq 1}\sum_{1\leq j\leq k_3}\frac{\lambda_i}{\mu_j}\left(\boldsymbol{v}_i^\top\boldsymbol{w}_j\right)^2.
\end{aligned}
\tag{49}
$$

Next we divide the above summation into $1 \leq i \leq k_1$ and $i > k_1$. For the first part, notice that

$$
\begin{aligned}
\sum_{1\leq j\leq k_3}\frac{\lambda_i}{\mu_j}\left(\boldsymbol{v}_i^\top\boldsymbol{w}_j\right)^2 &\leq \sum_{1\leq j\leq n}\frac{\lambda_i}{\mu_j}\left(\boldsymbol{v}_i^\top\boldsymbol{w}_j\right)^2 \\
&= \lambda_i\boldsymbol{v}_i^\top\left(\sum_{1\leq j\leq n}\frac{1}{\mu_j}\boldsymbol{w}_j\boldsymbol{w}_j^\top\right)\boldsymbol{v}_i \\
&= \lambda_i\boldsymbol{v}_i^\top\boldsymbol{W}\tilde{\boldsymbol{\Lambda}}^{-1}\boldsymbol{W}^\top\boldsymbol{v}_i \\
&= \lambda_i\boldsymbol{v}_i^\top\boldsymbol{W}\tilde{\boldsymbol{\Lambda}}^{\frac{1}{2}}\boldsymbol{U}^\top\boldsymbol{U}\tilde{\boldsymbol{\Lambda}}^{-2}\boldsymbol{U}^\top\boldsymbol{U}\tilde{\boldsymbol{\Lambda}}^{\frac{1}{2}}\boldsymbol{W}^\top\boldsymbol{v}_i \\
&= \lambda_i^2\tilde{\boldsymbol{x}}_i^\top\left(\boldsymbol{X}\boldsymbol{X}^\top\right)^{-2}\tilde{\boldsymbol{x}}_i,
\end{aligned}
\tag{50}
$$

where $\tilde{\boldsymbol{x}}_i$ is defined as $\tilde{\boldsymbol{x}}_i = \frac{\boldsymbol{X}\boldsymbol{v}_i}{\sqrt{\lambda_i}} = \frac{\boldsymbol{U}\tilde{\boldsymbol{\Lambda}}^{\frac{1}{2}}\boldsymbol{W}^\top\boldsymbol{v}_i}{\sqrt{\lambda_i}}$.

From the proof of lemma 11 in Bartlett et al. (2019), we know that for any $\sigma_x$, there exists a constant $c_0$ and $c$ such that if $k_0 \leq \frac{n}{c}$, with probability at least $1 - e^{-\frac{n}{c}}$ the first part can be bounded as

$$\sum_{1 \leq i \leq k_1} \sum_{1 \leq j \leq k_3} \frac{\lambda_i}{\mu_j} \left( \boldsymbol{v}_i^\top \boldsymbol{w}_j \right)^2 \leq \sum_{1 \leq i \leq k_1} \lambda_i^2 \tilde{\boldsymbol{x}}_i (\boldsymbol{X} \boldsymbol{X}^\top)^{-2} \tilde{\boldsymbol{x}}_i \leq c \frac{k_1}{n}, \tag{51}$$

which gives a bound for the first part.

For the second part we interchange the order of summation and have

$$\begin{aligned}
\sum_{i \geq k_1} \sum_{1 \leq j \leq k_3} \frac{\lambda_i}{\mu_j} \left( \boldsymbol{v}_i^\top \boldsymbol{w}_j \right)^2 &= \sum_{1 \leq j \leq k_3} \sum_{i \geq k_1} \frac{\lambda_i}{\mu_j} \left( \boldsymbol{v}_i^\top \boldsymbol{w}_j \right)^2 \\
&\leq \frac{1}{c_3 c(t,n) \sum_{i>0} \lambda_i} \sum_{1 \leq j \leq k_3} \sum_{i \geq k_1} \lambda_i \left( \boldsymbol{v}_i^\top \boldsymbol{w}_j \right)^2 \\
&= \frac{\lambda_{k_1+1}}{c_3 c(t,n) \sum_{i>0} \lambda_i} \sum_{1 \leq j \leq k_3} \sum_{i \geq k_1} \left( \boldsymbol{v}_i^\top \boldsymbol{w}_j \right)^2 \\
&\leq \frac{\lambda_{k_1+1}}{c_3 c(t,n) \sum_{i>0} \lambda_i} \sum_{1 \leq j \leq k_3} 1 \\
&= \frac{\lambda_{k_1+1} k_3}{c_3 c(t,n) \sum_{i>0} \lambda_i} \\
&\leq c \frac{k_3}{c(t,n)n}.
\end{aligned} \tag{52}$$

for $c$ large enough.

Putting 51 and 52 together, and noting that $k_3 \leq k_2$ with high probability as given in lemma A.8, we know there exists a constant $c$ that with probability at least $1 - e^{-\frac{n}{c}}$,

$$① \leq c \frac{k_1}{n} + c \frac{k_2}{c(t,n)n}. \tag{53}$$

**Bounding ②**

Similar to the first step in bounding ①, we note that

$$\begin{aligned}
② = \mathrm{Tr} &\left[ \boldsymbol{U} \mathrm{diag} \left\{ \overbrace{0, \cdots 0}^{k_3 \text{ times}}, \overbrace{\left[ 1 - \left( 1 - \frac{\lambda}{n} \mu_{k_3+1} \right)^t \right]^2, \cdots, \left[ 1 - \left( 1 - \frac{\lambda}{n} \mu_n \right)^t \right]^2}^{n-k_3 \text{ times}} \right\} \right. \\
&\left. \boldsymbol{U} \tilde{\boldsymbol{\Lambda}}^{-2} \boldsymbol{U}^\top \boldsymbol{U} \tilde{\boldsymbol{\Lambda}}^{\frac{1}{2}} \boldsymbol{W}^\top \boldsymbol{\Sigma} \boldsymbol{W} \tilde{\boldsymbol{\Lambda}}^{\frac{1}{2}} \boldsymbol{U}^\top \right] \\
= \mathrm{Tr} &\left[ \mathrm{diag} \left\{ \overbrace{0, \cdots 0}^{k_3 \text{ times}}, \overbrace{\frac{1}{\mu_{k_3+1}} \left[ 1 - \left( 1 - \frac{\lambda}{n} \mu_{k_3+1} \right)^t \right]^2, \cdots, \frac{1}{\mu_n} \left[ 1 - \left( 1 - \frac{\lambda}{n} \mu_n \right)^t \right]^2}^{n-k_3 \text{ times}} \right\} \right. \\
&\left. \boldsymbol{W}^\top \boldsymbol{\Sigma} \boldsymbol{W} \right].
\end{aligned} \tag{54}$$

From Bernoulli's inequality and the definition of $k_3$, for any $k_3 + 1 \leq j \leq n$, we have

$$\frac{1}{\mu_k} \left[ 1 - \left( 1 - \frac{\lambda}{n} \mu_k \right)^t \right]^2 \leq \frac{1}{\mu_k} \left( \frac{\lambda}{n} \mu_k t \right)^2 = \left( \frac{\lambda t}{n} \right)^2 \mu_k \leq c_3 \left( \frac{\lambda t}{n} \right)^2 c(t,n) \sum_{i>0} \lambda_i, \tag{55}$$

Hence,

$$② \leq c_3 c(t, n) \left(\frac{\lambda t}{n}\right)^2 \sum_{i>0} \lambda_i \, \mathrm{Tr}[\boldsymbol{W}^\top \boldsymbol{\Sigma} \boldsymbol{W}]$$

$$= c_3 c(t, n) \left(\frac{\lambda t}{n} \sum_{i>0} \lambda_i\right)^2. \tag{56}$$

**Putting things together**

From the bounds for ① and ② given above, we know that there exists a constant $c$ such that with probability at least $1 - e^{-\frac{n}{c}}$, the trace of the variance matrix $C$ has the following upper bound

$$\mathrm{Tr}[C] \leq c \left(\frac{k_1}{n} + \frac{k_2}{c(t, n)n} + c(t, n) \left(\frac{\lambda t}{n} \sum_{i>0} \lambda_i\right)^2\right). \tag{57}$$

$\square$

*Proof of theorem 5.1.* Lemma A.4, A.7 and Theorem A.1 gives the complete proof. Note that the high probability events in the proof are independent of the epoch number $t$, and this implies that the theorem holds uniformly for all $t \in \mathbb{N}$. $\square$

### A.6 PROOF OF COMPATIBILITY RESULTS

**Corollary A.2** (Corollary 5.1 restated). *Let Assumption 1, 2 and 3 hold. Fix a constant $c(t, n)$. Suppose $k_0 = O(n)$, $k_1 = o(n)$, $r(\boldsymbol{\Sigma}) = o(n)$, $\lambda = O\left(\frac{1}{\sum_{i>0} \lambda_i}\right)$. Then there exists a sequence of positive constants $\{\delta_n\}_{n \geq 0}$ which converge to 0, such that with probability at least $1 - \delta_n$, the excess risk is consistent for $t \in \left(\omega\left(\frac{1}{\lambda}\right), o\left(\frac{n}{\lambda}\right)\right)$, i.e.*

$$R(\boldsymbol{\theta}_t) = o(1).$$

*Furthermore, for any positive constant $\delta$, with probability at least $1 - \delta$, the minimal excess risk on the training trajectory can be bounded as*

$$\min_t R(\boldsymbol{\theta}_t) \lesssim \frac{\max\{\sqrt{r(\boldsymbol{\Sigma})}, 1\}}{\sqrt{n}} + \frac{\max\{k_1, 1\}}{n}.$$

*Proof.* According to Lemma A.7, with probability at least $1 - \frac{\delta_n}{2}$, the following inequality holds for all $t$:

$$B(\boldsymbol{\theta}_t) \lesssim \left(\frac{1}{\lambda t} + \max\left\{\sqrt{\frac{r(\boldsymbol{\Sigma})}{n}}, \frac{r(\boldsymbol{\Sigma})}{n}, \sqrt{\frac{\log(\frac{1}{\delta_n})}{n}}, \frac{\log(\frac{1}{\delta_n})}{n}\right\}\right). \tag{58}$$

If $\delta_n$ is chosen such that $\log \frac{1}{\delta_n} = o(n)$, we have that with probability at least $1 - \frac{\delta_n}{2}$, we have for all $t = \omega\left(\frac{1}{\lambda}\right)$:

$$B(\boldsymbol{\theta}_t) = o(1), \tag{59}$$

in the sample size $n$.

When $c(t, n)$ is a constant, we have $k_2 \leq k_1$ with high probability as given in lemma A.8. Therefore, according to Lemma A.4 and Theorem A.1, we know that if $\log \frac{1}{\delta_n} = O(n)$, with probability at least $1 - \frac{\delta_n}{2}$, the following bound holds for all $t$:

$$V(\boldsymbol{\theta}_t) \lesssim \log\left(\frac{1}{\delta_n}\right)\left(\frac{k_1}{n} + \frac{\lambda^2 t^2}{n^2}\right). \tag{60}$$

Since $k_1 = o(n)$, $t = o\left(\frac{n}{\lambda}\right)$, we have $\frac{k_1}{n} + \frac{\lambda^2 t^2}{n^2} = o(1)$. Therefore, there exists a mildly decaying sequence of $\delta_n$ with $\log\left(\frac{1}{\delta_n}\right)\left(\frac{k_1}{n} + \frac{\lambda^2 t^2}{n^2}\right) = o(1)$, i.e.,

$$V(\boldsymbol{\theta}_t) = o(1). \tag{61}$$

To conclude, $\delta_n$ can be chosen such that

$$\log\left(\frac{1}{\delta_n}\right) = \omega(1), \log\left(\frac{1}{\delta_n}\right) = O(n), \log\left(\frac{1}{\delta_n}\right) = O\left(\frac{1}{\frac{k_1}{n} + \frac{\lambda^2 t^2}{n^2}}\right), \tag{62}$$

and then with probability at least $1 - \delta_n$, the excess risk is consistent for all $t \in \left(\omega\left(\frac{1}{\lambda}\right), o\left(\frac{n}{\lambda}\right)\right)$:

$$R(\boldsymbol{\theta}_t) = B(\boldsymbol{\theta}_t) + V(\boldsymbol{\theta}_t) = o(1). \tag{63}$$

This completes the proof for the first claim. The second claim follows from Equation 58 and 60 by setting $t = \Theta\left(\frac{\sqrt{n}}{\lambda}\right)$. $\qquad\square$

**Lemma A.9** (Lemma 5.1 restated). *For any fixed (i.e. independent of sample size $n$) feature covariance $\boldsymbol{\Sigma}$ satisfying assumption 1, we have $k_1(n) = o(n)$.*

*Proof.* Suppose there exists constant $c$, such that $k_1(n) \geq cn$. By definition of $k_1$, we know that $\lambda_l \geq \frac{c_1 \sum_{i>0} \lambda_i}{n}$ holds for $1 \leq l \leq k_1(n)$. Hence we have

$$\sum_{l=\lfloor cn2^i \rfloor + 1}^{\lfloor cn2^{i+1} \rfloor} \lambda_l \gtrsim \frac{c_1 \sum_{i>0} \lambda_i}{n2^{i+1}} cn2^i \gtrsim \sum_{i>0} \lambda_i. \tag{64}$$

summing up all $l$ leads to a contradiction since $\sum_{i>0} \lambda_i < \infty$, which finishes the proof. $\qquad\square$

**Theorem A.2** (Theorem 4.1 restated). *Consider the overparameterized linear regression setting defined in section 4.1. Let Assumption 1,2 and 3 hold. Assume the learning rate satisfies $\lambda = O(\frac{1}{\text{Tr}(\boldsymbol{\Sigma})})$. Then under the condition that $k_0 = O(n), k_1 = o(n), r(\boldsymbol{\Sigma}) = o(n)$, gradient descent is compatible with overparameterized linear regression in the region $T_n = \left(\omega\left(\frac{1}{\lambda}\right), o\left(\frac{n}{\lambda}\right)\right)$, namely,*

$$\sup_{t \in T_n} R(\boldsymbol{\theta}_t) \xrightarrow{P} 0 \quad as \quad n \to \infty.$$

*Furthermore, if the feature dimension $p = \infty$ and the data distribution does not change with $n$, then the condition $k_0 = O(n)$ alone suffices for compatibility.*

*Proof.* According to Corollary 5.1 and Lemma 5.1, for any $\varepsilon > 0$, there exists $\{\delta_n\}_{n>0}$ and $N$ such that for any sample size $n > N$, we have

$$\Pr\left[\left|\sup_{t \in T_n} R(\boldsymbol{\theta}_t)\right| > \varepsilon\right] \leq \delta_n. \tag{65}$$

Let $n \to \infty$ shows that $\sup_{t \in T_n} R(\boldsymbol{\theta}_t)$ converges to 0 in probability, which completes the proof. $\quad\square$

# B  EXAMPLES AND DISCUSSIONS

## B.1  EXAMPLE DISTRIBUTIONS

We apply our bound in Corollary 5.1 to several examples. In each example, we show the data distribution, the time interval, and the corresponding generalization bound. These distributions are widely discussed in (Bartlett et al., 2019; Zou et al., 2021).

**Example B.1.** *Under the same conditions as Theorem 5.1, let $\boldsymbol{\Sigma}$ denote the feature covariance matrix. We show the following examples:*

1. **(Inverse Polynominal).** *If the spectrum of $\boldsymbol{\Sigma}$ satisfies*

$$\lambda_k = \frac{1}{k^\alpha},$$

*for some $\alpha > 1$, we derive that $k_0 = \Theta(n), k_1 = \Theta\left(n^{\frac{1}{\alpha}}\right)$. Therefore, $\min_t V(\boldsymbol{\theta}_t) = O\left(n^{\frac{1-\alpha}{\alpha}}\right)$ and*

$$\min_t R(\boldsymbol{\theta}_t) = O\left(n^{-\min\left\{\frac{\alpha-1}{\alpha}, \frac{1}{2}\right\}}\right).$$

2. **(Inverse Log-Polynominal).** *If the spectrum of $\Sigma$ satisfies*

$$\lambda_k = \frac{1}{k \log^\beta (k+1)},$$

*for some $\beta > 1$, we derive that $k_0 = \Theta\left(\frac{n}{\log n}\right)$, $k_1 = \Theta\left(\frac{n}{\log^\beta n}\right)$. Therefore, $\min_t V(\boldsymbol{\theta}_t) = O\left(\frac{1}{\log^\beta n}\right)$ and*

$$\min_t R(\boldsymbol{\theta}_t) = O\left(\frac{1}{\log^\beta n}\right).$$

3. **(Constant).** *If the spectrum of $\Sigma$ satisfies*

$$\lambda_k = \frac{1}{n^{1+\varepsilon}}, 1 \le k \le n^{1+\varepsilon},$$

*for some $\varepsilon > 0$, we derive that $k_0 = 0$, $k_1 = 0$. Therefore, $\min_t V(\boldsymbol{\theta}_t) = O\left(\frac{1}{n}\right)$ and*

$$\min_t R(\boldsymbol{\theta}_t) = O\left(\frac{1}{\sqrt{n}}\right).$$

4. **(Piecewise Constant).** *If the spectrum of $\Sigma$ satisfies*

$$\lambda_k = \begin{cases} \frac{1}{s} & 1 \le k \le s, \\ \frac{1}{d-s} & s+1 \le k \le d, \end{cases}$$

*where $s = n^r, d = n^q, 0 < r \le 1, q \ge 1$. We derive that $k_0 = n^r$, $k_1 = n^r$. Therefore, $\min_t V(\boldsymbol{\theta}_t) = O(n^{r-1})$ and*

$$\min_t R(\boldsymbol{\theta}_t) = O\left(n^{-\min\left\{1-r, \frac{1}{2}\right\}}\right).$$

### B.2 COMPARISONS WITH BENIGN OVERFITTING RESULTS

We summarize the results in Bartlett et al. (2019); Zou et al. (2021) and our results in Table 1, and provide a detailed comparison with them below.

**Comparison to Bartlett et al. (2019).** In this seminal work, the authors study the excess risk of the min-norm interpolator. As discussed before, the min-norm interpolator is the convergence point of gradient descent in the overparameterized linear regression setting. One of the main results in Bartlett et al. (2019) is providing a tight bound for the variance part[5] in excess risk as

$$V(\hat{\boldsymbol{\theta}}) = O\left(\frac{k_0}{n} + \frac{n}{R_{k_0}(\Sigma)}\right), \tag{66}$$

where $\hat{\boldsymbol{\theta}} = \boldsymbol{X}^\top (\boldsymbol{X} \boldsymbol{X}^\top)^{-1} \boldsymbol{Y}$ denotes the min-norm interpolator, and $R_k(\Sigma) = (\sum_{i>k} \lambda_i)^2 / (\sum_{i>k} \lambda_i^2)$ denote another type of effective rank.

By introducing the time factor, Theorem 5.1 improves over Equation (66) in at least two aspects. Firstly, Theorem 5.1 guarantees the consistency of the gradient descent dynamics for a broad range of step number $t$, while Bartlett et al. (2019) only study the limiting behavior of the dynamics of $t \to \infty$. Secondly, Theorem 5.1 implies that the excess risk of early stopping gradient descent solution can be much better than the min-norm interpolator. Compared to the bound in Equation (66), the bound in Corollary 5.1 (a.) replaces $k_0$ with a much smaller quantity $k_1$; and (b.) drops the second term involving $R_{k_0}(\Sigma)$. Therefore, we can derive a consistent bound for overparameterized linear regression for an early stopping solution, even though the excess risk of limiting point (min-norm interpolator) can be $\Omega(1)$ (See the first example in B.1). We can further derive a data-dependent time interval in which the bound is consistent, which cannot be directly obtained in Bartlett et al. (2019).

---

[5]We do not compare the bias component in the excess risk bound here since our bound for the bias component follows Bartlett et al. (2019).

**Comparison to Zou et al. (2021).** Zou et al. (2021) study a different setting, which focuses on the one-pass stochastic gradient descent solution of linear regression. The authors prove a bound for the excess risk as

$$R(\bar{\boldsymbol{\theta}}_t) = O\left(\frac{k_1}{n} + \frac{n\sum_{i>k_1}\lambda_i^2}{(\sum_{i>0}\lambda_i)^2}\right),\tag{67}$$

where $\bar{\boldsymbol{\theta}}_t$ denotes the parameter obtained using stochastic gradient descent (SGD) with constant step size at epoch $t$. Similar to our bound, Equation 67 also uses the effective dimension $k_1$ to characterize the variance term. However, we emphasize that Zou et al. (2021) derive the bound in a pretty different scenario from ours, which is one-pass SGD scenario. During the one-pass SGD training, we use a fresh data point to perform stochastic gradient descent in each epoch, and therefore they set $t = \Theta(n)$ by default. As a comparison, we apply the standard full-batch gradient descent, and thus the time can be more flexible. Besides, our results in Corollary 5.1 improve the bound in Equation (67) by dropping the second term. We refer to the third and fourth example in Example B.1 for more details, where[6] our bound outperforms Zou et al. (2021) when $\varepsilon < 1/2$ or $q < \min\{2 - r, 3/2\}$.

### B.3 COMPARISONS WITH STABILITY-BASED BOUNDS

In this section, we show that Theorem 5.1 gives provably better upper bounds than the stability-based method. We cite a result from Teng et al. (2021), which uses stability arguments to tackle overparameteried linear regression under similar assumptions.

**Theorem B.1** (modified from Theorem 1 in Teng et al. (2021)). *Under the overparameterized linear regression settings, assume that $\|\boldsymbol{x}\| \leq 1$, $|\varepsilon| \leq V$, $w = \frac{\boldsymbol{\theta}^{*,\top}\boldsymbol{x}}{\sqrt{\boldsymbol{\theta}^{*,\top}\boldsymbol{\Sigma}\boldsymbol{\theta}^*}}$ is $\sigma_w^2$-subgaussian. Let $B_t = \sup_{\tau \in [t]} \|\boldsymbol{\theta}_t\|$. the following inequality holds with probability at least $1 - \delta$:*

$$R(\boldsymbol{\theta}_t) = \tilde{O}\left(\max\{1, \boldsymbol{\theta}^{*,\top}\boldsymbol{\Sigma}\boldsymbol{\theta}^*\sigma_w^2, (V + B_t)^2\}\sqrt{\frac{\log(4/\delta)}{n}} + \frac{\|\boldsymbol{\theta}^*\|^2}{\lambda t} + \frac{\lambda t(V + B_t)^2}{n}\right).\tag{68}$$

Theorem B.1 applies the general stability-based results (Hardt et al., 2016; Feldman & Vondrák, 2019) in the overparameterized linear regression setting, by replacing the bounded Lipschitz condition with the bounded domain condition. A fine-grained analysis (Lei & Ying, 2020) may remove the bounded Lipschitz condition, but it additionally requires zero noise or decaying learning rate, which is different from our setting. We omit the excess risk decomposition technique adopted in Teng et al. (2021) for presentation clarity.

Theorem B.1 can not directly yield the stability argument in Theorem 4.1, since obtaining a high probability bound of $B_t$ requires a delicate trajectory analysis and is a non-trivial task. Therefore, data-irrelevant methods such as stability-based bounds can not be directly applied to our setting. Even if one can replace $B_t$ in Equation 68 with its expectation that is easier to handle (this modification will require adding concentration-related terms, and make the bound in Equation 68 looser), we can still demonstrate that Theorem 5.1 is tighter than the corresponding stability-based analysis by providing a lower bound on $\mathbb{E}[B_t^2]$, which will imply a lower bound on the righthand side of Equation 68.

**Theorem B.2.** *Let Assumption 1, 2, 3 holds. Suppose $\lambda = O\left(\frac{1}{\sum_{i>0}\lambda_i}\right)$. Suppose the conditional variance of the noise $\varepsilon|\boldsymbol{x}$ is lower bounded by $\sigma_\varepsilon^2$. There exists constant c, such that with probability at least $1 - ne^{-\frac{n}{c}}$, we have for $t = o(n)$,*

$$\mathbb{E}\|\boldsymbol{\theta}_t\|^2 = \Omega\left(\frac{\lambda^2 t^2}{n}\left(\sum_{i>k_0}\lambda_i\right)\right)\tag{69}$$

First we prove the following lemma, bounding the number of large $\mu_i$.

---

[6]Due to the bias term in Theorem 5.1, the overall excess risk bound cannot surpass the order $O(1/\sqrt{n})$, which leads to the cases that Zou et al. (2021) outperforms our bound. However, we note that such differences come from the intrinsic property of GD and SGD, which may be unable to avoid in the GD regimes.

**Lemma B.1.** *Suppose $t = o(n)$. Let $l$ denote the number of $\mu_i$, such that $\mu_i = \Omega\left(\frac{n}{t}\right)$. Then with probability at least $1 - ne^{-\frac{n}{c}}$, we have $l = O(t)$.*

*Proof.* According to Lemma A.1, we know that with probability at least $1 - ne^{-\frac{n}{c}}$, Equation 15 holds for all $0 \le k \le n - 1$. Conditioned on this, we have

$$\frac{n}{t}l \lesssim \sum_{k=1}^{l} \mu_i \lesssim \sum_{k=1}^{l}(\sum_{i \ge k} \lambda_i + \lambda_k n) \lesssim (l + n)\sum_{i > 0} \lambda_i \lesssim l + n. \tag{70}$$

Since $t = o(n)$, we have $l = O(t)$ as claimed. $\qquad\square$

We also need the result from Bartlett et al. (2019), which gives a lowerbound of $\mu_n$.

**Lemma B.2.** *(Lemma 10 in Bartlett et al. (2019)) For any $\sigma_x$, there exists a constant c, such that with probability at least $1 - e^{-\frac{n}{c}}$ we have,*

$$\mu_n \ge c\left(\sum_{i > k_0} \lambda_i\right). \tag{71}$$

We are now ready to prove Theorem B.2.

*Proof.* We begin with the calculation of $\|\boldsymbol{\theta}_t\|^2$. By Lemma A.2, the conditional unbiasedness of noise in Assumption 2 and the noise variance lower bound, we have

$$
\begin{aligned}
\mathbb{E}\|\boldsymbol{\theta}_t\|^2 &= \left\|\left(\boldsymbol{I} - \frac{\lambda}{n}\boldsymbol{X}^\top\boldsymbol{X}\right)^t(\boldsymbol{\theta}_0 - \boldsymbol{X}^\dagger\boldsymbol{Y}) + \boldsymbol{X}^\dagger Y\right\|^2 \\
&= \mathbb{E}\left\|\left(\boldsymbol{I} - \left(\boldsymbol{I} - \frac{\lambda}{n}\boldsymbol{X}^\top\boldsymbol{X}\right)^t\right)\boldsymbol{X}^\dagger(\boldsymbol{X}\boldsymbol{\theta}^* + \varepsilon)\right\|^2 \\
&= \mathbb{E}\left\|\left(\boldsymbol{I} - \left(\boldsymbol{I} - \frac{\lambda}{n}\boldsymbol{X}^\top\boldsymbol{X}\right)^t\right)\boldsymbol{X}^\dagger\boldsymbol{X}\boldsymbol{\theta}^*\right\|^2 + \mathbb{E}\left\|\left(\boldsymbol{I} - \left(\boldsymbol{I} - \frac{\lambda}{n}\boldsymbol{X}^\top\boldsymbol{X}\right)^t\right)\boldsymbol{X}^\dagger\varepsilon\right\|^2 \\
&\ge \mathbb{E}\left\|\left(\boldsymbol{I} - \left(\boldsymbol{I} - \frac{\lambda}{n}\boldsymbol{X}^\top\boldsymbol{X}\right)^t\right)\boldsymbol{X}^\dagger\varepsilon\right\|^2 \\
&= \mathbb{E}\,\mathrm{Tr}\left[\left(\boldsymbol{I} - \left(\boldsymbol{I} - \frac{\lambda}{n}\boldsymbol{X}^\top\boldsymbol{X}\right)^t\right)\boldsymbol{X}^\dagger\varepsilon\varepsilon^\top\boldsymbol{X}^{\dagger,\top}\left(\boldsymbol{I} - \left(\boldsymbol{I} - \frac{\lambda}{n}\boldsymbol{X}^\top\boldsymbol{X}\right)^t\right)\right] \\
&\ge \sigma_\varepsilon^2\mathbb{E}\,\mathrm{Tr}\left[\left(\boldsymbol{I} - \left(\boldsymbol{I} - \frac{\lambda}{n}\boldsymbol{X}^\top\boldsymbol{X}\right)^t\right)\boldsymbol{X}^\dagger\boldsymbol{X}^{\dagger,\top}\left(\boldsymbol{I} - \left(\boldsymbol{I} - \frac{\lambda}{n}\boldsymbol{X}^\top\boldsymbol{X}\right)^t\right)\right] \\
&= \sigma_\varepsilon^2\sum_{i=1}^{n}\frac{[1 - (1 - \frac{\lambda}{n}\mu_i)^t]^2}{\mu_i}.
\end{aligned}
\tag{72}
$$

When $\mu_i = o\left(\frac{n}{t}\right)$, we have

$$1 - (1 - \frac{\lambda}{n}\mu_i)^t = 1 - 1 + \frac{\lambda}{n}\mu_i t + O\left(\left(\frac{\lambda}{n}\mu_i t\right)^2\right) = \Theta\left(\frac{\lambda}{n}\mu_i t\right). \tag{73}$$

Plugging it into Equation 72 and then use Lemma B.1, B.2, we know that under the high probability event in Lemma B.1 and B.2,

$$\mathbb{E}\|\boldsymbol{\theta}_t\|^2 = \Omega\left((n - l)\frac{\lambda^2}{n^2}\mu_n t^2\right) = \Omega\left(\frac{\lambda^2}{n}\mu_n t^2\right) = \Omega\left(\frac{\lambda^2 t^2}{n}\left(\sum_{i > k_0} \lambda_i\right)\right) \tag{74}$$

$\qquad\square$

Therefore, the stability-based bound, i.e., the right hand side of Equation 68, can be lower bounded in expectation as $\Omega\left(\frac{\lambda^3 t^3}{n^2}\sum_{i>k_0}\lambda_i\right)$. This implies that the stability-based bound is vacuous when $t = \Omega\left(\frac{n^{\frac{2}{3}}\left(\sum_{i>k_0}\lambda_i\right)^{-\frac{1}{3}}}{\lambda}\right)$. Thus, stability-based methods will provably yield smaller compatibility region than $\left(\omega\left(\frac{1}{\lambda}\right), o\left(\frac{n}{\lambda}\right)\right)$ in Theorem 4.1 when $\sum_{i>k_0}\lambda_i$ is not very small, as demonstrated in the examples below.

**Example B.2.** *Let Assumption 1, 2, 3 holds. Assume without loss of generality that $\lambda = \Theta(1)$. We have the following examples:*

1. **(Inverse Polynominal).** *If the spectrum of $\Sigma$ satisfies*

$$\lambda_k = \frac{1}{k^\alpha},$$

   *for some $\alpha > 1$, we derive that $k_0 = \Theta(n)$, $\sum_{i>k_0}\lambda_i = \Theta(\frac{1}{n^{\alpha-1}})$. Therefore, the stability bound in Theorem B.1 is vacuous when*

$$t = \Omega\left(n^{\frac{\alpha+1}{3}}\right),$$

   *which is outperformed by the compatibility region in Theorem 5.1 when $\alpha < 2$.*

2. **(Inverse Log-Polynominal).** *If the spectrum of $\Sigma$ satisfies*

$$\lambda_k = \frac{1}{k\log^\beta(k+1)},$$

   *for some $\beta > 1$, we derive that $k_0 = \Theta\left(\frac{n}{\log n}\right)$, $\sum_{i>k_0}\lambda_i = \tilde{\Theta}(1)$. Therefore, the stability bound in Theorem B.1 is vacuous when*

$$t = \tilde{\Omega}\left(n^{\frac{2}{3}}\right),$$

   *which is outperformed by the compatibility region in Theorem 5.1.*

3. **(Constant).** *If the spectrum of $\Sigma$ satisfies*

$$\lambda_k = \frac{1}{n^{1+\varepsilon}}, 1 \le k \le n^{1+\varepsilon},$$

   *for some $\varepsilon > 0$, we derive that $k_0 = 0$, $\sum_{i>k_0}\lambda_i = 1$. Therefore, the stability bound in Theorem B.1 is vacuous when*

$$t = \Omega\left(n^{\frac{2}{3}}\right),$$

   *which is outperformed by the compatibility region in Theorem 5.1.*

4. **(Piecewise Constant).** *If the spectrum of $\Sigma$ satisfies*

$$\lambda_k = \begin{cases} \frac{1}{s} & 1 \le k \le s, \\ \frac{1}{d-s} & s+1 \le k \le d, \end{cases}$$

   *where $s = n^r, d = n^q, 0 < r \le 1, q \ge 1$, we derive that $k_0 = n^r$, $\sum_{i>k_0}\lambda_i = 1$. Therefore, the stability bound in Theorem B.1 is vacuous when*

$$t = \Omega\left(n^{\frac{2}{3}}\right),$$

   *which is outperformed by the compatibility region in Theorem 5.1.*

## B.4 COMPARISONS WITH UNIFORM CONVERGENCE BOUNDS

We give a standard bound on the Rademacher complexity of linear models.

**Theorem B.3** (Theorem in Mohri et al. (2012)). *Let $S \subseteq \{\boldsymbol{x} : \|x\|_2 \leq r\}$ be a sample of size $n$ and let $\mathcal{H} = \{x \mapsto \langle w, x \rangle : \|w\|_2 \leq \Lambda\}$. Then, the empirical Rademacher complexity of $\mathcal{H}$ can be bounded as follows:*

$$\hat{\mathcal{R}}_S(\mathcal{H}) \leq \sqrt{\frac{r^2 \Lambda^2}{n}}. \tag{75}$$

Furthermore, Talagrand's Lemma (See Lemma 5.7 in Mohri et al. (2012)) indicates that

$$\hat{\mathcal{R}}_S(l \circ \mathcal{H}) \leq L\hat{\mathcal{R}}_S(\mathcal{H}) = \frac{\Theta(\Lambda^2)}{\sqrt{n}}, \tag{76}$$

where $L = \Theta(\Lambda)$ is the Lipschitz coefficient of the square loss function $l$ in our setting. Therefore, the Rademacher generalization bound is vacuous when $\Lambda = \Omega(n^{\frac{1}{4}})$. By Theorem B.2, we know that $\mathbb{E}\|\boldsymbol{\theta}_t\|^2 = \Omega(n^{\frac{1}{2}})$ when $t = \Omega\left(\frac{n^{\frac{3}{4}}}{\lambda\left(\sum_{i>k_0}\lambda_i\right)^{\frac{1}{2}}}\right)$. A similar comparison as in Example B.2 can demonstrate that uniform stability arguments will provably yield smaller compatibility region than that in Theorem 5.1 for example distributions.

### B.5 COMPARISON WITH PREVIOUS WORKS ON EARLY STOPPING

A line of works focuses on deriving the excess risk guarantee of linear regression or kernel regression with early stopping (stochastic) gradient descent. We refer to Section 2 for a detailed discussion. Here we compare our results with some most relevant works, including (Yao et al., 2007; Lin & Rosasco, 2017; Pillaud-Vivien et al., 2018).

**Comparison with Yao et al. (2007).** Yao et al. (2007) study kernel regression with early stopping gradient descent and share some similarities with our paper. However, their approaches cannot cover ours, and are fundamentally different from ours in the following aspects.

Firstly, the assumptions used in the two approaches are different, due to different goals and techniques. Yao et al. (2007) assume that the input feature and data noise have bounded norm (see Section 2.1 Definitions and Notations in Yao et al. (2007)), while we require that the input feature is subgaussian with independent entries. The assumption used in our paper is widely-used in benign overfitting analysis, following Bartlett et al. (2019).

Furthermore, although Yao et al. (2007) obtain a minimax bound in terms of the convergence rate, it is suboptimal in terms of compatibility region. Specifically, The results in our paper show a compatibility region like $(0, n)$ while the techniques Yao et al. (2007) can only lead to a compatibility region like $(0, \sqrt{n})$. See Proof of the Main Theorem in section 2 in Yao et al. (2007) for details. Such differences come from different goals of the two approaches, where Yao et al. (2007) focus on providing the optimal early-stopping time while we focus on providing a larger compatibility region.

**Comparison with Pillaud-Vivien et al. (2018).** Pillaud-Vivien et al. (2018) study kernel regression with multi-pass stochastic gradient descent and derive optimal excess risk guarantee. Different from our approaches with full-batch gradient descent, they study *averaged* stochastic gradient descent with batchsize equal to 1.

**Comparison with Lin & Rosasco (2017).** Lin & Rosasco (2017) study stochastic gradient descent with arbitrary batchsize, which is reduced to full batch gradient descent when setting the batchsize to sample size $n$. However, their results are still fundamentally different from ours, since they require the boundness assumption, and focus more on the optimal early stopping time rather than the largest compatibility region, in the same spirit of Yao et al. (2007). Specifically, Lin & Rosasco (2017) derive a compatibility region like $(0, n^{\frac{\zeta+1}{2\zeta+\gamma}})$, where $\zeta$ and $\gamma$ are problem dependent constants (See Theorem 1 in Lin & Rosasco (2017) for details). The following examples demonstrate that this paper's results yield larger compatibility regions for a wide range of distribution classes.

**Example B.3. (Inverse Polynominal).** *If the spectrum of $\boldsymbol{\Sigma}$ satisfies*

$$\lambda_k = \frac{1}{k^\alpha},$$

*for some $\alpha > 1$. For this distribution, we have $\zeta = \frac{1}{2}$, $\gamma = \frac{1}{\alpha}$, and the compatibility region is $(0, n^{\frac{3\alpha}{2\alpha+1}})$, which is smaller than the compatibility region $(0, n^{\frac{3\alpha+1}{2\alpha+1}})$ given in Example B.5.*

**Example B.4. (Inverse Log-Polynominal).** *If the spectrum of $\boldsymbol{\Sigma}$ satisfies*

$$\lambda_k = \frac{1}{k \log^\beta (k+1)},$$

*for some $\beta > 1$. For this distribution, we have $\zeta = \frac{1}{2}$, $\gamma = 1$, and the compatibility region is $(0, n^{\frac{3}{4}})$, which is smaller than the compatibility region $(0, n)$ given Corollary 5.1.*

### B.6 DISCUSSIONS ON EXTENSIONS TO KERNEL REGRESSION

In this section, we discuss extending the analysis in overparameterized linear regression to kernel regression setting.

Let $\mathcal{H}$ denote a infinite dimensional Hilbert space equipped with inner product $\langle \cdot, \cdot \rangle_{\mathcal{H}}$, and $\phi : \mathbb{R}^d \to \mathcal{H}$ denote a feature map. Consider the following class of functions:

$$\mathcal{F} = \{f : \mathbb{R}^d \to \mathbb{R} | f(\boldsymbol{x}) = \langle \boldsymbol{\theta}, \phi(\boldsymbol{x}) \rangle_{\mathcal{H}}\}. \tag{77}$$

Let $(\boldsymbol{x}, y) \in \mathbb{R}^d \times \mathbb{R}$ denote the data vector and the response, following a joint distribution $\mathcal{D}$. The goal of kernel regression is to find a function $f$ parameterized by $\boldsymbol{\theta}$, that minimizes the following population loss

$$L(\boldsymbol{\theta}) = \frac{1}{2} \mathbb{E} \left(y - f(\boldsymbol{x})\right)^2 = \frac{1}{2} \mathbb{E} \left(y - \langle \boldsymbol{\theta}, \phi(\boldsymbol{x}) \rangle_{\mathcal{H}}\right)^2 \tag{78}$$

Therefore, kernel regression is equivalent to solving a linear regression problem on transformed data $(\phi(\boldsymbol{x}), y)$. By replacing $\boldsymbol{x}$ with $\phi(\boldsymbol{x})$, the notations and results in Section 4 can be naturally extended to the kernel regression case, which is detailed as follows.

Given a dataset $\{(\boldsymbol{x}_i, y_i)\}_{1 \le i \le n}$ sampled i.i.d from $\mathcal{D}$, consider the following dynamics of gradient descent analogous to Equation (3),

$$\boldsymbol{\theta}_{t+1} = \boldsymbol{\theta}_t - \frac{\lambda}{n} \phi(\boldsymbol{X})^\top (\phi(\boldsymbol{X}) \boldsymbol{\theta}_t - \boldsymbol{Y}), \tag{79}$$

where $\phi(\boldsymbol{X}) = (\phi(\boldsymbol{x}_1), \cdots, \phi(\boldsymbol{x}_n))^\top$ and $\boldsymbol{Y} = (y_1, \cdots, y_n)^\top$. Note that this dynamic takes place in the feature space $\mathcal{H}$. The following corollary characterizes the compatibility between kernel regression and gradient descent.

**Corollary B.1.** *Assume the distribution of $(\phi(\boldsymbol{x}), y)$ satisfies Assumptions 1, 2, 3, and does not change with sample size $n$. Let $\boldsymbol{\Sigma}_\phi = \mathbb{E} \left[\phi(\boldsymbol{x})\phi(\boldsymbol{x})^\top\right]$ denote the feature covariance. Then under the condition that the effective dimension $k_0(\boldsymbol{\Sigma}_\phi) = O(n)$ and the learning rate $\lambda = O(\frac{1}{\mathrm{Tr}(\boldsymbol{\Sigma}_\phi)})$, gradient descent using 79 is compatible with kernel regression.*

*Proof.* The corollary follows from Theorem 4.1, by noting that its proof holds for feature vectors in an infinite dimensional Hilbert space. $\square$

We give several remarks regarding the difference between the generalization analysis of kernel regression and overparameterized linear regression.

Firstly, Corollary B.1 assumes $\phi(\boldsymbol{x})$ has subgaussian i.i.d entries after normalization (see Assumption 1.3), which can be hard to satisfy due to the non-linearity of feature mapping $\phi(\cdot)$.

Secondly, Corollary B.1 focuses on gradient descent in the feature space $\mathcal{H}$. Another perspective is to apply the Representer theorem (Mohri et al., 2012) to express the weight $\boldsymbol{\theta}$ into a linear combination of transformed inputs as $\boldsymbol{\theta} = \sum_{1 \le j \le n} \alpha_j \phi(\boldsymbol{x}_j)$, and analyze the gradient descent dynamics of the $n$-dimensional vector $\boldsymbol{\alpha} = (\alpha_1, \cdots, \alpha_n)$. Although they both converge to the min-norm solution, these two gradient descent dynamics are different, and Theorem 4.1 is not directly applicable to gradient descent on $\boldsymbol{\alpha}$.

Thirdly, previous works (El Karoui, 2010; Liang & Rakhlin, 2018; Mei & Montanari, 2022) prove that the spectrum of feature covariance $\boldsymbol{\Sigma}_\phi$ has an approximately linear relationship with the spectrum of data covariance $\boldsymbol{\Sigma} = \mathbb{E} \left[\boldsymbol{x}\boldsymbol{x}^\top\right]$ when feature mapping $\phi(\cdot)$ corresponds to an inner product type kernel. This can be used to derive a more intrinsic compatibility condition using only data distribution rather than feature distribution. We leave a more refined analysis for data-algorithm compatibility in the kernel regression regime for future works.

### B.7 VARYING $t$, VARYING $c(t, n)$

Although setting $c(t, n)$ to a constant as in Corollary 5.1 suffices to prove Theorem 4.1, in this section we show that the choice of $c(t, n)$ can be much more flexible. Specifically, we provide a concrete example and demonstrate that by setting $c(t, n)$ to a non-constant, Theorem 5.1 can indeed produce larger compatibility regions.

**Example B.5.** *Under the same conditions as Theorem 5.1, let $\boldsymbol{\Sigma}$ denote the feature covariance matrix. If the spectrum of $\boldsymbol{\Sigma}$ satisfies $\lambda_k = \frac{1}{k^\alpha}$ for some $\alpha > 1$, we set $c(t, n) = \Theta\left(n^{\frac{\alpha+1-2\alpha\tau}{2\alpha+1}}\right)$ for a given $\frac{\alpha+1}{2\alpha} \le \tau \le \frac{3\alpha+1}{2\alpha+2}$. Then for $t = \Theta(n^\tau)$, we derive that $V(\boldsymbol{\theta}_t) = O\left(n^{\frac{2\alpha\tau-3\alpha+2\tau-1}{2\alpha+1}}\right)$.*

Example B.5 shows that by choosing $c(t, n)$ as a non-constant, we exploit the full power of Theorem 5.1, and extend the compatibility region to $t = o\left(n^{\frac{3\alpha+1}{2\alpha+2}}\right) = \omega(n)$. In this example, Theorem 5.1 outperforms all $O\left(\frac{t}{n}\right)$-type bounds, which become vacuous when $t = \omega(n)$.

### B.8 CALCULATIONS IN SECTION B.1

We calculate the quantities $r(\boldsymbol{\Sigma}), k_0, k_1, k_2$ for the example distributions in B.1. The results validate that $k_1$ is typically a much smaller quantity than $k_0$, and $k_1$ serves as a proxy for $k_2$ in the constant $c(t, n)$ setting.

1. **Calculations for $\lambda_k = \frac{1}{k^\alpha}, \alpha > 1$.**

   Define $r_k(\boldsymbol{\Sigma}) = \frac{\sum_{i>k} \lambda_i}{\lambda_{k+1}}$ as in Bartlett et al. (2019). Since $\sum_{i>k} \frac{1}{i^\alpha} = \Theta(\frac{1}{k^{\alpha-1}})$, we have $r_k(\boldsymbol{\Sigma}) = \Theta\left(\frac{\frac{1}{k^{\alpha-1}}}{\frac{1}{k^\alpha}}\right) = \Theta(k)$. Hence, $k_0 = \Theta(n)$ [7], and the conditions of theorem 5.1 is satisfied.

   As $\sum_{i>0} \lambda_i < \infty$, By its definition we know that $k_1$ is the smallest $l$ such that $\lambda_{l+1} = O(\frac{1}{n})$. Therefore, $k_1 = \Theta(n^{\frac{1}{\alpha}})$. Similarly, $k_2 = \Theta(n^{\frac{1}{\alpha}})$.

2. **Calculations for $\lambda_k = \frac{1}{k \log^\beta(k+1)}, \beta > 1$.**

   $\sum_{i>k} \frac{1}{i \log^\beta(i+1)} = \Theta(\int_k^\infty \frac{1}{x \log^\beta x} dx) = \Theta(\frac{1}{\log^{\beta-1} k})$, which implies $r_k(\boldsymbol{\Sigma}) = k \log k$. Solving $k_0 \log k_0 \ge \Theta(n)$, we have $k_0 = \Theta(\frac{n}{\log n})$.

   By the definition of $k_1$, we know that $k_1$ is the smallest $l$ such that $l \log^\beta(l+1) \ge \Theta(n)$. Therefore, $k_1 = \Theta(\frac{n}{\log^\beta n})$. $k_2 = \Theta(\frac{n}{\log^\beta n})$ by similar calculations.

3. **Calculations for $\lambda_i = \frac{1}{n^{1+\varepsilon}}, 1 \le i \le n^{1+\varepsilon}, \varepsilon > 0$.**

   Since $r_0(\boldsymbol{\Sigma}) = n^{1+\varepsilon}$, we have $k_0 = 0$. By the definition of $k_1$ and $k_2$, we also have $k_1 = k_2 = 0$.

4. **Calculations for $\lambda = \begin{cases} \frac{1}{s} & 1 \le k \le s \\ \frac{1}{d-s} & s+1 \le k \le d \end{cases}, s = n^r, d = n^q, 0 < r \le 1, q \ge 1$.**

   For $0 \le k < s, r_k(\boldsymbol{\Sigma}) = \Theta(\frac{1}{\frac{1}{s}}) = \Theta(n^r) = o(n)$, while $r_s(\boldsymbol{\Sigma}) = \frac{1}{\frac{1}{d-s}} = \Theta(n^q) = \omega(n)$. Therefore, $k_0 = s = n^r$.

   Similarly, noting that $\lambda_k = \frac{1}{n^r} = \omega(n)$ for $0 \le k < s$ and $\lambda_s = \Theta(\frac{1}{n^d}) = o(n)$, we know that $k_1 = s = n^r$. Similarly, $k_2 = s = n^r$.

### B.9 CALCULATIONS FOR $\lambda_k = \frac{1}{k^\alpha}, \alpha > 1$ IN SECTION B.7

Set $c(t, n) = \frac{1}{n^\beta}$, where $\beta > 0$ will be chosen later. First we calculate $k_2$ under this choice of $c(t, n)$. Note that $\sum_{i>k} \frac{1}{i^\alpha} = \Theta\left(\frac{1}{k^{\alpha-1}}\right)$. Therefore, $k_2$ is the smallest $k$ such that $\frac{1}{k^{\alpha-1}} + \frac{n}{k^\alpha} = O(\frac{1}{n^\beta})$. For the bound on $V(\theta_t)$ to be consistent, we need $k_2 = o(n)$. Hence, $\frac{1}{k^{\alpha-1}} = O(\frac{n}{k^\alpha})$, which implies $k_2 = n^{\frac{\beta+1}{\alpha}}$.

---

[7] The calculations for $k_0, k_1$ and $k_2$ in this section only apply when $n$ is sufficiently large.

Plugging the value of $c(t, n)$ and $k_2$ into our bound, we have

$$V(\theta_t) = O\left(n^{(\frac{1}{\alpha}-1)+(\frac{1}{\alpha}+1)\beta} + n^{2\tau-\beta-2}\right)$$

which attains its minimum $\Theta(n^{\frac{2\alpha\tau-3\alpha+2\tau-1}{2\alpha+1}})$ at $\beta = \Theta\left(\frac{2\alpha\tau-\alpha-1}{2\alpha+1}\right)$.

For $V(\theta) = O(1)$, we need $\tau \leq \frac{3\alpha+1}{2\alpha+2}$. For $\beta \geq 0$, we need $\tau \geq \frac{\alpha+1}{2\alpha}$. Putting them together gives the range of $t$ in which the above calculation applies.

## B.10 DISCUSSION ON $\mathcal{D}_n$

In this paper, the distribution $\mathcal{D}$ is regarded as a sequence of distribution $\{\mathcal{D}_n\}$ which may dependent on sample size $n$. The phenomenon comes from overparameterization and asymptotic requirements. In the definition of compatibility, we require $n \to \infty$. In this case, overparameterization requires that the dimension $p$ (if finite) cannot be independent of $n$ since $n \to \infty$ would break overparameterization. Therefore, the covariance $\Sigma$ also has $n$-dependency, since $\Sigma$ is closely related to $p$.

Several points are worth mentioning: (1) Similar arguments generally appear in related works. For example, Bartlett et al. (2019) use similar arguments when discussing the definition of benign covariance (Page 7 in the arxiv version). (2) One can avoid such issues by letting $p \to \infty$. This is why we discuss the special case $p = \infty$ in Theorem 4.1. (3) If $p$ is a fixed finite constant that does not alter with $n$, the problem becomes underparameterized and thus trivial to get a consistent generalization bound.

## B.11 COMPARISONS WITH PAC-BAYES BOUNDS

As one of the most exciting techniques in generalization analysis, PAC-Bayes theory works both theoretically and empirically. Usually, the form of PAC-Bayes bounds relies on the distance between the *prior distribution* which is unrelated to the training procedure, and the *posterior distribution* which is the distribution after training (*e.g.*, isotropic Gaussian distribution centered at a trained parameter).

In this sense, PAC-Bayes considers different regimes from this paper, since the returned classifier in this paper is not a distribution. Moreover, PAC-Bayes usually do not explicitly focus on early-stopping iterates, while the notion of compatibility heavily relies on the early-stopping arguments. By considering early-stopping that encodes algorithm information, we can derive tighter generalization bound relying on weaker assumptions.

*More interestingly, PAC-Bayes framework is not mutually exclusive with the trajectory analysis.* One can indeed introduce trajectory analysis into PAC-Bayes techniques and derive a compatibility region based on trajectory-based PAC-Bayes theory. By doing so, one can explicitly incorporate more algorithm information into PAC-Bayes framework. We leave more detailed discussions for future work.

We next show the differences in more detail:

1. **Different settings.** In PAC-Bayes analysis, the returned classifier is a distribution instead of a fixed parameter. Forcing PAC-Bayes analysis in the fixed parameter regime would cost much because it is hard to define distribution distance (*e.g.*, KL divergence) on two *single-point distributions*.

2. **Different classifiers.** Although both methods consider "a bag of" classifiers, they are fundamentally different. In PAC-Bayes framework, the trained random classifiers are regarded to be *independently* drawn from the posterior distribution. However, in trajectory analysis, all classifiers are *dependent* during the training process. Therefore, trajectory analysis is more challenging in this sense.

3. **Different characterizations.** PAC-Bayes framework characterizes the *expectation* of generalization loss, over the randomness of the posterior distribution over parameters, which do not explicitly sketch the time factor. In comparison, the trajectory analysis in this paper focus on providing a *compatibility region*, where the generalization error is uniformly consistent and considers the time factor.

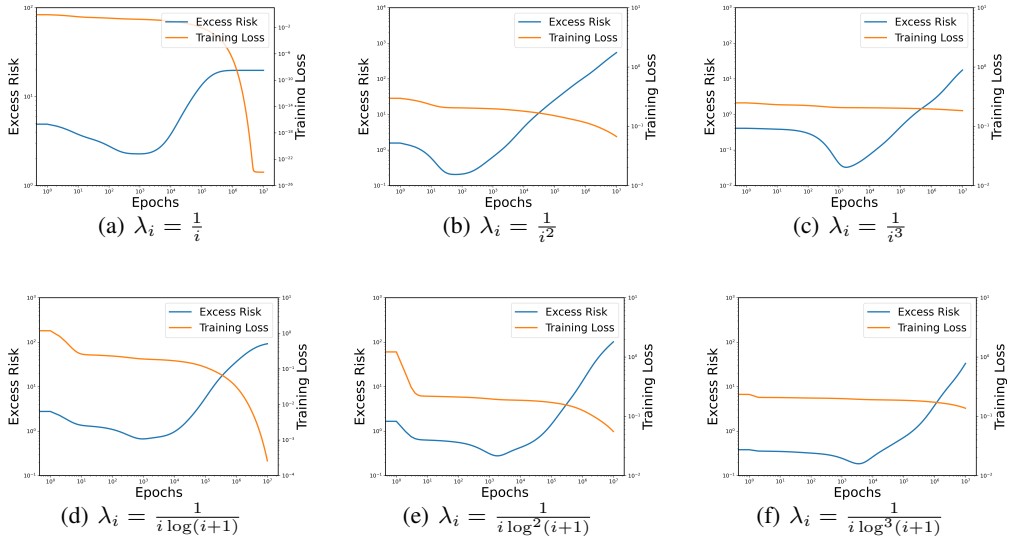

Figure 2: **The training plot for overparameterized linear regression with different covariances using GD.**

## C ADDITIONAL EXPERIMENT RESULTS

### C.1 DETAILS AND DISCUSSIONS FOR LINEAR REGRESSION EXPERIMENTS

In this section, we provide the experiment details for linear regression experiments and present additional empirical results.

In Section 6, We consider six overparameterized linear regression instances with input dimension $p = 1000$ and sample size $n = 100$. The feature vectors are independently sampled from zero-mean Gaussian distributions, whose covariances are diagonal with entries $\lambda_i = \frac{1}{i}, \frac{1}{i^2}, \frac{1}{i^3}, \frac{1}{i\log(i+1)}, \frac{1}{i\log^2(i+1)}, \frac{1}{i\log^3(i+1)}$, respectively. For each feature $\boldsymbol{x}$, we construct the response as $y = \boldsymbol{x}^\top \boldsymbol{\theta}^* + \varepsilon$, where $\boldsymbol{\theta}^*$ is sampled from a $p$-dimensional standard Gaussian distribution for each instance, and $\varepsilon$ is sampled from a standard Gaussian distribution. We conduct gradient descent with learning rate $\lambda = 0.001$ on the above instances. We also compute min-norm excess risk via its closed form, with $1 \times 10^{-4}$ weight decay on parameter $\boldsymbol{\theta}$ to avoid numerical instability. Note that the regularization will only reduce the final excess and will not jeopardize our final conclusion's correctness.

The linear regression experiment in Figure 1 follows the setting described in section 6. Although the final iterate does not interpolate the training data, the results suffice to demonstrate the gap between the early-stopping and final-iterate excess risk. The training plot for different covariances are given in Figure 2.

We provide the experiment results of sample sizes $n = 50$, $n = 200$ and feature dimensions $p = 500$, $p = 2000$ analogous to those in Section 6. The settings are the same as described in the main text, except for the sample size. The optimal excess risk and min-norm excess risk are provided in Table 3, 4, 6 and 5. The tables indicate that the three observations given in section 6 hold for different sample size $n$.

### C.2 DETAILS FOR MNIST EXPERIMENTS

In this section, we provide the experiment details and additional results in MNIST dataset.

The MNIST experiment details are described below. We create a noisy version of MNIST with label noise rate $20\%$, i.e. randomly perturbing the label with probability $20\%$ for each training data,

Table 3: **The effective dimension $k_1$, the optimal early stopping excess risk and min-norm excess risk for different feature distributions, with sample size $n = 50$, $p = 1000$.** We calculate the 95% confidence interval for the excess risk.

| DISTRIBUTIONS | $k_1$ | OPTIMAL EXCESS RISK | MIN-NORM EXCESS RISK |
|---|---|---|---|
| $\lambda_i = \frac{1}{i}$ | $\Theta(n)$ | $2.515 \pm 0.0104$ | $12.632 \pm 0.1602$ |
| $\lambda_i = \frac{1}{i^2}$ | $\Theta(n^{\frac{1}{2}})$ | $0.269 \pm 0.0053$ | $50.494 \pm 0.9378$ |
| $\lambda_i = \frac{1}{i^3}$ | $\Theta(n^{\frac{1}{3}})$ | $0.083 \pm 0.0011$ | $13.208 \pm 0.4556$ |
| $\lambda_i = \frac{1}{i \log(i+1)}$ | $\Theta(\frac{n}{\log n})$ | $0.808 \pm 0.0090$ | $46.706 \pm 0.6983$ |
| $\lambda_i = \frac{1}{i \log^2(i+1)}$ | $\Theta(\frac{n}{\log^2 n})$ | $0.381 \pm 0.0076$ | $74.423 \pm 1.1472$ |
| $\lambda_i = \frac{1}{i \log^3(i+1)}$ | $\Theta(\frac{n}{\log^3 n})$ | $0.233 \pm 0.0052$ | $43.045 \pm 0.8347$ |

Table 4: **The effective dimension $k_1$, the optimal early stopping excess risk and min-norm excess risk for different feature distributions, with sample size $n = 200$ , $p = 1000$.** We calculate the 95% confidence interval for the excess risk.

| DISTRIBUTIONS | $k_1$ | OPTIMAL EXCESS RISK | MIN-NORM EXCESS RISK |
|---|---|---|---|
| $\lambda_i = \frac{1}{i}$ | $\Theta(n)$ | $2.173 \pm 0.0065$ | $52.364 \pm 0.4009$ |
| $\lambda_i = \frac{1}{i^2}$ | $\Theta(n^{\frac{1}{2}})$ | $0.161 \pm 0.0039$ | $36.855 \pm 0.4833$ |
| $\lambda_i = \frac{1}{i^3}$ | $\Theta(n^{\frac{1}{3}})$ | $0.068 \pm 0.0012$ | $8.1990 \pm 0.2313$ |
| $\lambda_i = \frac{1}{i \log(i+1)}$ | $\Theta(\frac{n}{\log n})$ | $0.628 \pm 0.0034$ | $152.70 \pm 1.1073$ |
| $\lambda_i = \frac{1}{i \log^2(i+1)}$ | $\Theta(\frac{n}{\log^2 n})$ | $0.241 \pm 0.0036$ | $83.550 \pm 0.7596$ |
| $\lambda_i = \frac{1}{i \log^3(i+1)}$ | $\Theta(\frac{n}{\log^3 n})$ | $0.146 \pm 0.0108$ | $33.469 \pm 0.4540$ |

to simulate the label noise which is common in real datasets, e.g ImageNet (Stock & Cissé, 2018; Shankar et al., 2020; Yun et al., 2021). We do not inject noise into the test data.

We choose a standard four layer convolutional neural network as the classifier. We use a vanilla SGD optimizer without momentum or weight decay. The initial learning rate is set to $0.5$. Learning rate is decayed by $0.98$ every epoch. Each model is trained for 300 epochs. The training batch size is set to 1024, and the test batch size is set to 1000. We choose the standard cross entropy loss as the loss function.

We provide the plot for different levels of label noise in Figure 3. We present the corresponding test error of the best early stopping iterate and the final iterate in Table 7. Since the theoretical part of this paper focuses on GD, we also provide a corresponding plot of GD training in Figure 4 for completeness.

Table 5: **The effective dimension $k_1$, the optimal early stopping excess risk and min-norm excess risk for different feature distributions, with sample size $n = 100$, $p = 500$.** We calculate the 95% confidence interval for the excess risk.

| DISTRIBUTIONS | $k_1$ | OPTIMAL EXCESS RISK | MIN-NORM EXCESS RISK |
|---|---|---|---|
| $\lambda_i = \frac{1}{i}$ | $\Theta(n)$ | $1.997 \pm 0.0876$ | $27.360 \pm 0.3019$ |
| $\lambda_i = \frac{1}{i^2}$ | $\Theta(n^{\frac{1}{2}})$ | $0.211 \pm 0.0050$ | $43.531 \pm 0.7025$ |
| $\lambda_i = \frac{1}{i^3}$ | $\Theta(n^{\frac{1}{3}})$ | $0.076 \pm 0.0011$ | $10.062 \pm 0.3022$ |
| $\lambda_i = \frac{1}{i \log(i+1)}$ | $\Theta(\frac{n}{\log n})$ | $0.645 \pm 0.0056$ | $96.465 \pm 1.0594$ |
| $\lambda_i = \frac{1}{i \log^2(i+1)}$ | $\Theta(\frac{n}{\log^2 n})$ | $0.289 \pm 0.0055$ | $83.694 \pm 0.9827$ |
| $\lambda_i = \frac{1}{i \log^3(i+1)}$ | $\Theta(\frac{n}{\log^3 n})$ | $0.181 \pm 0.0045$ | $38.090 \pm 0.6378$ |

Table 6: **The effective dimension $k_1$, the optimal early stopping excess risk and min-norm excess risk for different feature distributions, with sample size $n = 100$, $p = 2000$.** We calculate the 95% confidence interval for the excess risk.

| DISTRIBUTIONS | $k_1$ | OPTIMAL EXCESS RISK | MIN-NORM EXCESS RISK |
|---|---|---|---|
| $\lambda_i = \frac{1}{i}$ | $\Theta(n)$ | $2.662 \pm 0.0066$ | $23.111 \pm 0.2278$ |
| $\lambda_i = \frac{1}{i^2}$ | $\Theta(n^{\frac{1}{2}})$ | $0.219 \pm 0.0050$ | $43.130 \pm 0.6421$ |
| $\lambda_i = \frac{1}{i^3}$ | $\Theta(n^{\frac{1}{3}})$ | $0.077 \pm 0.0010$ | $10.031 \pm 0.2942$ |
| $\lambda_i = \frac{1}{i \log(i+1)}$ | $\Theta(\frac{n}{\log n})$ | $0.749 \pm 0.0055$ | $88.744 \pm 0.9414$ |
| $\lambda_i = \frac{1}{i \log^2(i+1)}$ | $\Theta(\frac{n}{\log^2 n})$ | $0.312 \pm 0.0057$ | $82.859 \pm 0.9394$ |
| $\lambda_i = \frac{1}{i \log^3(i+1)}$ | $\Theta(\frac{n}{\log^3 n})$ | $0.190 \pm 0.0047$ | $37.782 \pm 0.5945$ |

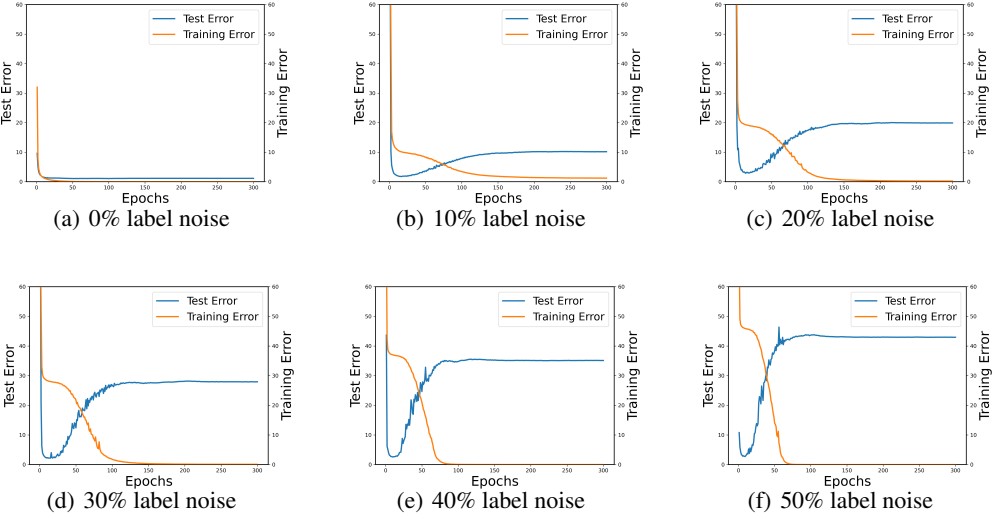

Figure 3: **The training plot for corrupted MNIST with different levels of label noise using SGD.** Figure (c) is copied from Figure 1.

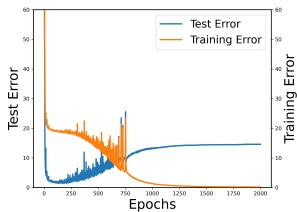

Figure 4: **The training plot for corrupted MNIST with 20% label noise using GD.**

Table 7: **The test error of optimal stopping iterate and final iterate on MNIST dataset with different levels of label noise.** The results demonstrate that stopping iterate can have significantly better generalization performance than interpolating solutions for real datasets.

| NOISE LEVEL | OPTIMAL TEST ERROR | FINAL TEST ERROR |
|---|---|---|
| 0% | 1.07% | 1.13% |
| 10% | 1.75% | 10.16% |
| 20% | 2.88% | 19.90% |
| 30% | 2.18% | 27.94% |
| 40% | 2.57% | 35.15% |
| 50% | 2.71% | 42.95% |

