# OpenReview forum: "When Do Models Generalize? A Perspective From Data-Algorithm Compatibility"
_ICLR.cc/2023/Conference — Submitted to ICLR 2023_

### Official Review · Reviewer_rKAH · 2022-10-19

**Confidence:** 3
**Clarity, Quality, Novelty And Reproducibility:** My overall comments are done above.
**Correctness:** 3
**Technical Novelty And Significance:** 2
**Empirical Novelty And Significance:** Not applicable
**Recommendation:** 5

**Strength And Weaknesses:**

### **Strength**

The authors introduce a framework that can be rich theoretically. Instead of studying the *implicit bias* of an algorithm at convergence, it studies whether the path of training dynamics has reached a good generalization point in a certain time zone. They also prove a separation with benign overfitting under certain condition on the spectrum of the covariance matrix.

### **Weaknesses**

Despite the fact that the set-up is clear, the (only) reel results presented are the one of gradient descent for a linear model. The link between early stopping and good generalization properties has already be well studied and I am surprised the authors cite Yao et al, Ali et al as they are doing exactly the same thing! As far as I understood, the only difference is the way to present the results: here, as now commonly stated in the benign overfitting, the generalization bounds are presented under the form of certain cut-off eigenvalue summations. Whereas in Yao et al, eigenvalue decay as polynomials are stated to make the bounds more readable. Morevover, on the contrary to what the authors say, the Assumption 1 *is not mild* (mainly the independence of the entries of $\tilde{x}$). As the rates of Yao et al are known to be minimax in the setup of the paper, I do not think that the present analysis could be tighter than theirs.


**Minor issues**

- Prefer *at convergence* than *the last iterate*
- Need a deeper comparison with early stopping.
- Under this form, Assumption 1 is hard to understand: the notation $O, o, w$ are not defined: are they with respect to $n$? If this is so, how to understand $o(n/\lambda)$, does $\lambda$ depends on $n$, if yes, $\lambda = O(1/Tr(\Sigma))$ is misleading ?

**Summary Of The Paper:**

The authors present a compatibility condition between the data distribution and the algorithm used for regression. This corresponds to a weaker requirement than the benign overfitting studies that simply study the generalization property *at convergence*.

**Summary Of The Review:**

Without a proper comparison with the papers from early stopping, I cannot consider the restatement of it under *compatibility condition* as a proper contribution. I am happy to change my evaluation if I am wrong.

---

> ### Author Response · Authors · 2022-11-09
> **Clarifications**
>
> We thank the reviewer for the constructive review and suggestions. We revise the paper accordingly, which is marked by **blue**. Below, we do our best to address your questions adequately.
>
> >Q1: The results are linear model [...] Yao et al, Ali et al as they are doing exactly the same thing! [...] the Assumption~1 is not mild due to the independence of the entries of x [...]  As the rates of Yao et al are known to be minimax in the setup of the paper, I do not think that the present analysis could be tighter than theirs.
>
> A1: We thank the reviewer for the insightful question. As the reviewer claimed, Yao et al., Ali et al., etc., also focus on the early-stopping settings. However, their approaches cannot cover ours, and are fundamentally different from ours in the following aspects:
>
> (1) ***Their results cannot cover ours***. As an example, although Yao et al. obtain a minimax bound in terms of the convergence rate, it is suboptimal in terms of compatibility region, i.e., the time interval in which the excess risk is consistent. Speficially, The results in our paper show a compatibility region like $(0, n)$ while the techniques in Yao et al. can only lead to a compatibility region like $(0, \sqrt{n})$ (see the following for more details). Such differences come from different goals of the two approaches, where Yao et al. focus on providing the optimal early-stopping time while we focus on providing a larger compatibility region.
>
> (Detailded Comparison) The bound in Yao et al. performs like (see Proof of the Main Theorem in section 2 in [1], right above Section 3)
> $$ \frac{t^{1-\theta}}{\sqrt{n}}+...,$$
> where $n$ is the sample size, $\theta$ is a constant related to the step size decay (in our setting, $\theta = 0$). Therefore, the bound becomes vacuous when $t=\Omega(\sqrt{n})$. By comparison, we can guarantee that the excess risk is $o(1)$ up to $t=o(n)$ for a $\Theta(1)$ stepsize (see corollary 5.1 ). Therefore, our bound makes improvements over theirs in terms of compatibility region.
>
> (2) The assumptions used in the two approaches are different, due to different goals and techniques. Yao et al. assume that the input feature and data noise have bounded norm (see Section 2.1 Definitions and Notations in [1], the paragraph before Remark 2.1), while we require that the input feature is subGaussian with independent entries after normalization. The assumption used in our paper is widely-used in benign overfitting analysis, following Bartlett et al. We revise the term "mild" to "widely-used" in the new version.
>
> (3) One of the points of this paper is to formalize the algorithm information in generalization analysis using early-stopping arguments, which emphasizes the critical role of early-stopping in generalization analysis. After all, we only train a model with finite steps in most practical cases.
>
> We add a more thorough discussion in the new version (see ***Section 2 and Appendix B.5***), including the related work and the comparison of the results. Thanks for the helpful advice.
>
> >Q2: Prefer at convergence than the last iterate.
>
> A2: Thanks for the suggestion, and we have made the adjustment.
>
> >Q3: Need a deeper comparison with early stopping.
>
> A3: We have added a detailed comparison with early stopping in Appendix B.5, and cite the related paper in the related works section.
>
> >Q4: Under this form, Assumption 1 is hard to understand: the notation $O,o,\omega$ are not defined: are they with respect to $n$? If this is so, how to understand $o(n/\lambda)$, does $\lambda$ depends on $n$, if yes, $\lambda=O(1/Tr(\Sigma))$ is misleading ?
>
> A4: We thank the reviewer for the valuable comment. In our paper, $O,o,\omega$ are defined with respect to sample size $n$. Note that the learning rate $\lambda$ can depend on $n$, since we allow the data distribution (and also $O(1/Tr(\Sigma))$) to vary with $n$.
>
>
> We appreciate the reviewer's comments that will help us shape discussion points. We look forward to any further discussions that would help the evaluation.
>
> [1] Yao, Y., Rosasco, L., and Caponnetto, A. (2007). On early stopping in gradient descent learning.

---

> > ### Comment · Reviewer_rKAH · 2022-11-09
> > **Short question to be clear.**
> >
> > Hence, if I understand properly, when you write $\lambda = O(1/tr(\Sigma))$, it is because you implicitly assume $tr(\Sigma) \to \infty$ when $n \to \infty$ ? If this is true, is the fact that the **population** covariance matrix goes to $\infty$ common here (where I come from there is no sense of making this $n$-dependence of the covariance, but I remain open-minded ) ? Why ? Can you provide more modelling assumption to justify this ?

---

> > > ### Author Response · Authors · 2022-11-10
> > > **Further Clarifications**
> > >
> > > We apologize for the confusion here. This argument does not imply that $tr(\Sigma) \to \infty$ as $n \to \infty$.
> > >
> > > The argument $\lambda=O(1/tr(\Sigma))$ means that there exists a large enough constant (which does not change with $n$), such that $$\lambda\le \frac{c}{tr(\Sigma)},$$
> > >  for any sample size $n\in \mathbb{N}_+$. The asymptotic notation does not imply that either side has to go to $0$ or $\infty$ necessarily. For example, when the trace term $tr(\Sigma)$ is a constant, both the step size $\lambda = c^\prime/\sqrt{n}$ (decay with n) and $\lambda = c^\prime$ (constant step size) are valid, where $c^\prime$ is a small enough constant. We have revised our manuscript accordingly (See the Notations paragraph in Section 4.1).
> > >
> > > We thank the reviewer again for the nice comments. We are happy to provide any further clarification to help with the evaluation.

---

> > > > ### Comment · Reviewer_rKAH · 2022-11-10
> > > > **Clarifications**
> > > >
> > > > I am sorry, of course I miswrite: I wanted to say, that when you write $O(1/tr(\Sigma))$, you implicitly assume that $tr(\Sigma)$ *can* depend on $n$, and forr example either go to $+\infty$, or be bounded. Is that right ?
> > > > To be clearer once again, it is *unusual*, at least to me, that the **population covariance** has a $n$-dependency! $n$ is a quantity related to the samples you have, whether $\Sigma$ is the underlying distribution of the samples...
> > > >
> > > >  So why do you assume this ?

---

> > > > > ### Author Response · Authors · 2022-11-10
> > > > > **Thanks for the question**
> > > > >
> > > > > We apologize for the confusion. The unusual phenomenon comes from overparameterization and asymptotic requirements.
> > > > >
> > > > > In the definition of compatibility, we require $n \to \infty$.
> > > > > In this case, overparameterization requires that the dimension $p$ (if finite) cannot be independent of $n$ since $n \to \infty$ would break overparameterization.
> > > > > Therefore, the covariance $\Sigma$ also has $n$-dependency, since $\Sigma$ is closely related to $p$.
> > > > > As mentioned in Def 3.1 (compatibility), the distribution $D$ in this paper should be regarded as a sequence of distributions $D_n$, due to the above discussion.
> > > > > We have added Appendix B.10 (Page 32, marked in blue) in the revision for discussion.
> > > > >
> > > > > Several points are worth mentioning:
> > > > > 1. Similar arguments appear in related works. For example, Bartlett et al. use similar arguments when discussing the definition of benign covariance (Page 7 in the arxiv version).
> > > > > 2. One can avoid such issues by letting $p = \infty$. This is why we discuss the special case $p = \infty$ in Theorem 4.1.
> > > > > 3. If $p$ is a fixed finite constant that does not alter with n, the problem becomes underparameterized, and thus is trivial to get a consistent generalization bound.
> > > > >
> > > > > Again, we thank the reviewer for the nice comments! We are eager to engage in further discussions to clear out any confusion.
> > > > >
> > > > > [1] Benign Overfitting in Linear Regression, Peter L. Bartlett, Philip M. Long, Gabor Lugosid, and Alexander Tsiglera.

---

> > > > > > ### Comment · Reviewer_rKAH · 2022-11-11
> > > > > > **Alright**
> > > > > >
> > > > > > Alright! Then I understand, thanks for the clarification (maybe I read it too fast, but I believe this needs to be said clearly at some point).
> > > > > >
> > > > > > Then I guess that, indeed the authors may be right about the kernel set up ($p = \infty$) being not representative of their study.
> > > > > > I remain pretty surprised that gradient descent bounds for the linear problem are not tight (even for this new criterion): hence, I would very like the authors to make a better comparison with the literature: e.g. Yao et al. is an old article and I believe that more recent bounds may be tighter. For example am thinking of **Optimal Rates for Multi-pass Stochastic Gradient Methods** from *Junhong Lin*, *Lorenzo Rosasco* or **Statistical Optimality of Stochastic Gradient Descent on Hard Learning Problems through Multiple Passes**, *Loucas Pillaud-Vivien*, *Alessandro Rudi*, *Francis Bach* that have tight bound in the kernel case for gradient descent.
> > > > > >
> > > > > > That being said, I will lower my certitude about the work, and tend to be more neutral as it appears that I misread the paper.

---

> > > > > > > ### Author Response · Authors · 2022-11-11
> > > > > > > **Thank you!**
> > > > > > >
> > > > > > > We thank the reviewer for the constructive suggestions. We revise our paper in the following two aspects:
> > > > > > >
> > > > > > > 1. We discuss more related literature of this line [1-9] in related work section (early-stopping paragraph).
> > > > > > > 2. We provide more comparison results and show that their results cannot cover ours in terms of compatibility region in Appendix B.5.
> > > > > > >
> > > > > > > If there are any potentially missing related works, please point them out! We are happy to discuss them which would help enhance our manuscript considerably.
> > > > > > >
> > > > > > > We would like to thank the reviewer again for taking our response into consideration.
> > > > > > >
> > > > > > > [1] Yuan Yao, Lorenzo Rosasco, and Andrea Caponnetto. On early stopping in gradient descent learning.
> > > > > > >
> > > > > > > [2] Junhong Lin and Lorenzo Rosasco. Optimal rates for multi-pass stochastic gradient methods.
> > > > > > >
> > > > > > > [3] Loucas Pillaud-Vivien, Alessandro Rudi, and Francis R. Bach. Statistical optimality of stochastic gradient descent on hard learning problems through multiple passes.
> > > > > > >
> > > > > > > [4] Lorenzo Rosasco and Silvia Villa. Learning with incremental iterative regularization.
> > > > > > >
> > > > > > > [5] Aymeric Dieuleveut and Francis Bach. Nonparametric stochastic approximation with large step-sizes.
> > > > > > >
> > > > > > > [6] Gilles Blanchard and Nicole Kr ̈amer. Convergence rates of kernel conjugate gradient for random design regression.
> > > > > > >
> > > > > > > [7] L. Lo Gerfo, Lorenzo Rosasco, Francesca Odone, Ernesto De Vito, and Alessandro Verri. Spectral algorithms for supervised learning.
> > > > > > >
> > > > > > > [8] Junhong Lin and Volkan Cevher. Optimal rates for spectral-regularized algorithms with least-squares regression over hilbert spaces.
> > > > > > >
> > > > > > > [9] Pierre Tarres and Yuan Yao. Online learning as stochastic approximation of regularization paths: Optimality and almost-sure convergence.

---

> > > > > > > > ### Author Response · Authors · 2022-11-17
> > > > > > > > **Follow Up**
> > > > > > > >
> > > > > > > > We would like to express our sincere gratitude for the reviewer's constructive suggestions and comments. We have added the literature review and detailed comparisons in the new version, and we are eager to provide any further clarifications and discussions to help the evaluation. If we have successfully addressed the concerns, we would sincerely appreciate an increased score, if possible.

---

### Official Review · Reviewer_kDAx · 2022-10-19

**Confidence:** 3
**Clarity, Quality, Novelty And Reproducibility:** see comments in the "strength and wea…
**Correctness:** 3
**Technical Novelty And Significance:** 3
**Empirical Novelty And Significance:** Not applicable
**Recommendation:** 6

**Strength And Weaknesses:**

Strength:

Compared to the benign over-fitting study (Bartlett et. al. 2019), this work requires weaker conditions on the covariance matrix. Particularly, it allows the scenarios where the eigenvalues decay polynomially, $\lambda_k \sim 1/k^\alpha$. Instead,  (Bartlett et. al. 2019) requires a heavy tail of the spectrum of the covariance.

For the scenarios that are included by the conditions of this paper but are excluded by (Bartlett et. al. 2019), this paper obtains a theoretical guarantee of the existence of good generalization solutions during the GD training. However, the tradeoff for these weaker conditions is that the good generalization is not necessarily at the final iteration.

Weaknesses:

The analysis is still restricted to the simple linear regression model, while most of the interests of the machine learning community are on the non-linear models, especially neural networks. It is widely believed that many properties of, or finding/observations on, linear models do not extend to non-linear models.


Figure 1 is not consistent with the theoretical results and could be misleading.

>First, Figure 1b is trained using SGD, while the theory (Theorem 4.1) is on full-batch GD. These two algorithms do not have the same trajectory, hence, the U-shape curve for SGD in Figure 1b does not necessarily tell the same story as Theorem 4.1. A curve with full-batch GD training is needed in Figure 1.

>Second, Figure 1a seems still at the early stage of training, as the training loss is still large. It is hard to see whether the test loss will have a second descent. For overparameterized models, the training loss can touch zero, and we should train the model after the zero training loss is obtained.



**Summary Of The Paper:**

This paper analyzes the generalization performance of a simple model — linear regression. Given that a prior work (Bartlett et. al. 2019) has shown a benign over-fitting of the final iterate of SGD for over-parameterized linear models, this work with milder conditions theoretically shows that good generalization can also be achieved during the training, not necessarily by the last iterate.

**Summary Of The Review:**

The analysis in the paper requires weaker conditions and includes more scenairos, e.g., polynomically decaying eigenvalues of covariance, than prior works. For these scenarios, it proves existence of good generalization solutions during full-batch GD training.

However, the analysis is only performed on the simple linear regression model, and it is not clear whether it applies to non-linear models.

Some of the numerical verifications are not consistent with the theory.

---

> ### Author Response · Authors · 2022-11-09
> **Clarifications**
>
> We would like to thank the reviewer for the positive and helpful review. We revise the paper accordingly, which is marked by **blue**. Below, we do our best to address the  questions adequately.
>
>
> >Q1: The analysis is still restricted to the simple linear regression model, while most of the interests of the machine learning community are on the non-linear models, especially neural networks. It is widely believed that many properties of, or finding/observations on, linear models do not extend to non-linear models.
>
> A1: Thanks for the comment. In this paper, the overparameterized linear regression analysis serves as a preliminary but important case study for analyzing compatbility.
> Studying overparameterized linear regression is important for the following reasons.
>
> Firstly, analyzing the generalization behavior of overparameterized linear regression is a non-trivial and a not-completely-solved task, which has received a lot of attention in the generalization community in recent years [1,2,3].
>
> Secondly, although not all the phenomenon observed in nonlinear regimes can be proved using linear models, researchers still use linear models as a starting point to explain some of the phenomenon. For example, [1] studies benign overfitting in linear regimes. Our MNIST results in Figure 1b and Appendix C.2 also illustrate that the benefits of early stopping exist for more complex models and data distributions.
>
> Thirdly, overparemterized linear regression is closed related to modern machine learning models via the Neural Tangent Kernel framework, and we provide a kernel regression version of the main theorem in Appendix B.6 to show the variability of our results.
>
>
> >Q2: Figure 1 is not consistent with the theoretical results and could be misleading.
> First, Figure 1b is trained using SGD, while the theory (Theorem 4.1) is on full-batch GD. These two algorithms do not have the same trajectory, hence, the U-shape curve for SGD in Figure 1b does not necessarily tell the same story as Theorem 4.1. A curve with full-batch GD training is needed in Figure 1.
>
> A2: Thanks for the helpful advice. We provide the corresponding GD trained MNIST with 20% label noise in figure 4 in appendix C.2. The figure illustrates that the early stopping iterates on the GD trajactory has lower excess risk than that of the last iterate, which accords with SGD results. The GD and SGD experiments on MNIST demonstrates that the benefits of early stopping exists empirically for a wide range of training algorithms.
>
> >Q3: Figure 1a seems still at the early stage of training, as the training loss is still large. It is hard to see whether the test loss will have a second descent. For overparameterized models, the training loss can touch zero, and we should train the model after the zero training loss is obtained.
>
> A3: Thanks for the constructive suggestion. We train the model longer ($10^8$ epochs) and revise Figure 1a. The training loss decreases below $10^{-4}$, and we do not observe double descent phenomenon.
>
>
> We would like to thank the reviewer again for the helpful review. We look forward to any further discussions that would help your evaluation.
>
>
> [1]P. L. Bartlett, P. M. Long, G. Lugosi, and A. Tsigler. Benign overfitting in linear regression.
>
> [2]A. Tsigler and P. L. Bartlett. Benign overfitting in ridge regression.
>
> [3]D. Zou, J. Wu, V. Braverman, Q. Gu, and S. M. Kakade. Benign overfitting of constant-stepsize SGD for linear regression.

---

> > ### Comment · Reviewer_kDAx · 2022-11-15
> > **reply**
> >
> > I thank the authors for giving detailed answers for my questions. It addressed most of my concerns, especially the ones regarding Figure 1a and 1b.
> >
> > However, as raised by Reviewer rKAH, there is a lot similarity with the paper by Yao et al. Although the authors clarified that the results of this submission is not covered by [Yao et al] and it allows a wider compatibility region than [Yao et al], I feel this result is incremental and less amazing than I thought.

---

> > > ### Author Response · Authors · 2022-11-16
> > > **Thanks for the nice reply!**
> > >
> > > Thanks for the nice reply!
> > >
> > > We would like to clarify that the main body of this work is to introduce a novel notion of compatibility, which borrows the idea of early stopping into generalization analysis and captures both data and algorithm information. This is fundamentally different from the existing papers[1,2,3], which focus on the optimal rate / stopping point of early stopping.
> > >
> > > Technically, our results consistently outperform theirs in terms of compatibility regions. Specifically, we obtain a compatibility region of $(0,n)$, while [1] can only get $(0,n^{\frac{1}{2}})$ and [2] can only get $(0,n^{\frac{3}{4}})$ (See Appendix B.5 for a detailed comparison). This is not an incremental improvement. The improvement comes from that we consider different settings and use different techniques compared with existing works [1,2,3]. Therefore, the contributions in our manuscript are non-trivial and distinct from previous works. In this sense, it is unfair to say it is incremental, which is a big criticism.
> > >
> > > We would like to provide any further clarifications to eliminate the reviewer's concerns.
> > >
> > > [1] Yao, Y., Rosasco, L., and Caponnetto, A. (2007). On early stopping in gradient descent learning.
> > >
> > > [2] Junhong Lin and Lorenzo Rosasco. Optimal rates for multi-pass stochastic gradient methods.
> > >
> > > [3] Loucas Pillaud-Vivien, Alessandro Rudi, and Francis R. Bach. Statistical optimality of stochastic gradient descent on hard learning problems through multiple passes.

---

### Official Review · Reviewer_9FRm · 2022-10-25

**Confidence:** 3
**Correctness:** 3
**Technical Novelty And Significance:** 4
**Empirical Novelty And Significance:** Not applicable
**Recommendation:** 6

**Clarity, Quality, Novelty And Reproducibility:**

The paper is novel. The results are stated clearly.
I've not gone through the proofs in detail; I did go through the key lemma statements in the appendix and how they connect to each other.

**Strength And Weaknesses:**

Strengths
===
1. The paper makes a valuable extension over the existing studies of overparameterized linear regression. While earlier studies look at the min-norm interpolator, this paper goes a step further to analyze intermediate solutions, which is highly non-trivial and interesting.
2. Early-stopping is also a practically relevant concept, thus justifying the significance of the setting considered in the paper.
3. It is also interesting that they are able to formally identify scenarios where the final iterate has overfitting that is _not_ benign, while the early-stopped solution does have excess risk converging to zero (which they term as "compatibility"). The formal notion of compatibility itself is novel.
4. The proof seems to be a non-trivial extension of Bartlett et al., 2019.
4.  The paper is well-written. The proof and its implications were clearly outlined in the main paper. There are also many example distributions which are helpful in drawing comparisons with benign overfitting.

Weaknesses
===
1. My main complaint is with the way the result is framed. I am afraid I do not agree with the motivation/narrative that this paper analyzes the interplay between data and algorithm while prior work. While classical bounds (such as VC dimensions), do ignore the algorithm altogether, more recent bounds (such as those based on norms, flatness etc.,), _do_ care about how the algorithm induces various biases in the model. Even in the context of studies in high-dimensional linear regression, the very fact that they study the min-norm interpolator says that they are looking at an algorithm-dependent solution and not any ERM solution. What might be correct to say here is that while previous studies of overparameterized linear regression perform a trajectory-independent analysis, this paper takes a trajectory-dependent analysis.

Overall, I would frame this paper as "a statistical study of early-stopping"  (which _is_ valuable in itself) rather than the current claim of  a "capturing data-algorithm interplay which has never been done before" which seems misguided and disproportionate.

2. I'd like to map the result intuitively to the motivation behind early-stopping, such as the one outlined in the introduction. Early-stopping helps the classifier ignore noise in the data, which in this case should correspond to the variance term. Is this essentially the reason why the early-stopping iterate's bound is better than the final iterate? e.g., in Thm 4.1, does $V(\theta) \neq o(1)$ when  $\theta$ is the final iterate?
Adding a discussion of how the result maps to the standard intuition of early-stopping would be helpful.

### Minor points:
3. Since the order of quantifiers is very important (the $\forall t$ must follow _after_ the "w.h.p"), I'd be explicit about this order in all the lemmas and intermediate theorems.

4. Sec B.4 argues that uniform convergence upper bounds do not enjoy as strong a guarantee. I wonder if it's possible to show a lower bound on uniform convergence such as in the sense of Negrea et al., 2020 i.e., consider the set of early-stopped iterates for various draws of the data, and show that the uniform convergence bound considered on that set has poor sample-complexity dependence.

5. I'm not sure I followed the first step in Eq 7. Minor: might be worth labeling every equation.


**Summary Of The Paper:**

- This paper provides a follow-up of the "benign overfitting" result from  Bartlett et al., 2019.
- While the benign overfitting setting studies the min-norm interpolator of the data (which corresponds to the _final_ solution learned by linear regression), this paper considers excess risk guarantees for intermediate early-stopped solutions of linear regression. In this sense, the paper analyzes the trajectory of GD solutions.
- This allows them to prove convergence of excess risk for a much more general class of data distributions than was covered in Bartlett et al., 2019.


**Summary Of The Review:**

The paper presents an interesting, significant and non-trivial extension of the statistical analsyses in overparameterized linear regression to early-stopped settings.  The proof and results are explained well. There are issues with how the motivation is framed, which I hope can be addressed during the rebuttal.

### Update after response

Thanks to the authors for their response. I appreciate the new added comparisons to related work.

I had concerns that the claim suggesting that this is the first technique that is data- and algorithm- dependent is too strong.
The authors admit that there are indeed existing techniques that depend on data and algorithm, but that this dependence is too superficial.
While I agree, I still believe that centering the paper around the notion of "compatibility" as a way of encoding algorithm and data-dependence is not as valuable and significant in itself as the paper claims. The paper does make a solid contribution in terms of statistically analyzing early-stopping for the benign overfitting type of setting; however, it also claims that there's novel insight and value in the definition of compatibility, which I do not see. Therefore, I wish to keep my score as it is. I strongly encourage the authors to rethink the story of the paper and focus on highlighting the key technical contributions rather than the abstraction of compatibility --- unless, there are multiple applications of compatibility beyond linear regression that can somehow drive home the point that this abstraction is insightful.

---

> ### Author Response · Authors · 2022-11-09
> **Clarifications (1)**
>
> We thank the reviewer for the detailed, helpful and professional review! We have found the comments very constructive! We revise the paper accordingly, which is marked by **blue**. Below, we do our best to address the questions.
>
> >Q1: My main complaint is with the way the result is framed. I am afraid I do not agree with the motivation/narrative that this paper analyzes the interplay between data and algorithm while prior work. While classical bounds (such as VC dimensions), do ignore the algorithm altogether, more recent bounds (such as those based on norms, flatness etc.,), do care about how the algorithm induces various biases in the model. Even in the context of studies in high-dimensional linear regression, the very fact that they study the min-norm interpolator says that they are looking at an algorithm-dependent solution and not any ERM solution. What might be correct to say here is that while previous studies of overparameterized linear regression perform a trajectory-independent analysis, this paper takes a trajectory-dependent analysis. Overall, I would frame this paper as "a statistical study of early-stopping" (which is valuable in itself) rather than the current claim of a "capturing data-algorithm interplay which has never been done before" which seems misguided and disproportionate.
>
>
> A1：Thanks for the insightful question. Indeed, previous generalization methods(e.g., Rademacher complexity) also incorporates different forms of algorithm information (e.g., parameter norm, min-norm solution) However, they only returns large generalization error [1,2].
> For example, we show in Appendix that the existing approaches (e.g., parameter norm) returns compatibility region smaller than ours. This comes from the fact, these methods encode the entire algorithm information into a single term, which may lose much algorithm information.
> The taxonomy of data-dependent / algorithm-dependent technique does not mean that they totally ignore all the information of algorithm/data, but mean that it loses some important algorithm/data information which makes it vacuous/inconsistent in generalization analysis.
> We have revised our manuscript accordingly (See Footnote 1). Please refer to the second paragraph of the general response for details.
>
> >Q2: I'd like to map the result intuitively to the motivation behind early-stopping, such as the one outlined in the introduction. Early-stopping helps the classifier ignore noise in the data, which in this case should correspond to the variance term. Is this essentially the reason why the early-stopping iterate's bound is better than the final iterate? e.g., in Thm 4.1, does $V(\theta)\neq o(1)$ when $\theta$ is the final iterate? Adding a discussion of how the result maps to the standard intuition of early-stopping would be helpful.
>
> A2: Exactly! The greatest significance of early stopping, at least in the overparameterized linear regression setting, is that the classifier will not overfit to the noise. As shown in the proof, we split the covariance into the leading eigenspace and the tailing eigenspace. The eigenvalues in the tailing part will cause the variance term in the excess risk of interpolating solutions to be $\Omega(1)$ for fast decaying spectrum, as is the case in [3]. However, since the convergence in the tailing eigenspace is slower compared with the leading eigenspace, a proper early stopping strategy will prevent the overfitting in the tailing eigenspace and meanwhile avoid underfitting in the leading eigenspace. This is a high level intuition of why early stopping works in our setting, and we have added the discussions in Section 5.1.

---

> > ### Author Response · Authors · 2022-11-09
> > **Clarifications (2)**
> >
> > >Q3: Since the order of quantifiers is very important (the ∀t must follow after the "w.h.p"), I'd be explicit about this order in all the lemmas and intermediate theorems.
> >
> > A3: Thanks for pointing this out. We have revised the maunscript accordingly.
> >
> > >Q4: Sec B.4 argues that uniform convergence upper bounds do not enjoy as strong a guarantee. I wonder if it's possible to show a lower bound on uniform convergence such as in the sense of Negrea et al., 2020 i.e., consider the set of early-stopped iterates for various draws of the data, and show that the uniform convergence bound considered on that set has poor sample-complexity dependence.
> >
> > A4: Currently, we are not able to directly apply the techniques in [4] to prove an $\Omega(1)$ lower bound for early stopping iterates. The major reason is that the proof in [2] is based on a noise-flipping argument, and this requires the classifier has enough expressive power to completely fit the data noise, which may not hold for the early stopping iterates. Therefore, the techniques in [4] can not be directly applied to our setting.
> >
> > As a surrogate, we prove a norm-based lower bound in Appendix B.4, but we believe that estabilishing a tighter lower bound is an interesting and worthwhile task, and we leave it for future work.
> >
> > >Q5: I'm not sure I followed the first step in Eq 7. Minor: might be worth labeling every equation.
> >
> > A5: We are sorry for the confusion here. The largest eigenvalue $\mu_1$ of $X^\top X$ corresponds to the smallest eigenvalue $1-\frac{\lambda}{n}\mu_1$ of $I-\frac{\lambda}{n}X^\top X$. We have rewrite the corresponding parts to make it more clear, and relable the equations in the manuscript.
> >
> > Once again, we appreciate the time the reviewer took to provide us with a detailed and constructive review! Please let us know if any further clarification is needed.
> >
> > [1] Zhang, C., Bengio, S., Hardt, M., Recht, B., and Vinyals, O. (2021). Understanding deep learning (still) requires rethinking generalization.
> >
> > [2] Nagarajan, V. and Kolter, J. Z. (2019). Uniform convergence may be unable to explain generalization in deep learning.
> >
> > [3] A. Tsigler and P. L. Bartlett. Benign overfitting in ridge regression.
> >
> > [4] Negrea, J., Dziugaite, G. K., and Roy, D. (2020). In defense of uniform convergence: Generalization via derandomization with an application to interpolating predictors.

---

### Official Review · Reviewer_a4RZ · 2022-10-28

**Confidence:** 3
**Correctness:** 3
**Technical Novelty And Significance:** 3
**Empirical Novelty And Significance:** 3
**Recommendation:** 6

**Clarity, Quality, Novelty And Reproducibility:**

For the details of these points, please see my review of "Strength and Weakness". However, to give a high-level answer:

**Clarity:** The paper is not very clear. Sometimes it is hard to follow.

**Quality:** The proofs and statements are strong and have high quality.

**Novelty:** The paper might have some novelty, but I am not experienced in generalization theory enough to judge.

**Reproducibility:** This work would be considered as a "theory" work where the findings are clear, hence I believed can be implemented by researchers if they find this work relevant.

**Strength And Weaknesses:**

I would like to note that, I am **not** an expert in generalization. I am aware of the general idea, but I would not consider myself as a researcher who is up to date with all the advancements. However, I would have been definitely interested in reading this work if I were to see this paper in the proceedings, hence I will ask questions that I am genuinely wondering, and I will specify strengths and weaknesses from my perspective.

**Strength:** The assumptions are mostly standard and I cannot foresee an "assumption-related problem". I really like Theorem 1 and its proof looks correct as much as I am concerned. Corollary 5.1 might have a good impact. Appendix B is extremely well, but "too well" that some of the results can even move back to the main paper (e.g., without seeing Example B.1 it is hard to imagine the main idea of the relevant Thm; similarly, Theorem B.2. looks very interesting).

**Weaknesses:**
*I list the major and minor concerns that made me give a "marginal accept" instead of "accept". However, I still think the paper has some novelty hence I am also going to stay active during the discussion period in case the authors have updates or questions regarding my review.*

***Major Weaknesses/Questions***
- The paper after Definition 3.1 is more or less all about linear regression with gradient descent. The "strong" proofs exploit a lot about the structure of linear regression, which already has lots of cool generalization bounds in the literature. I am therefore wondering, to what extent can we use it for other models. Because if we already know we want to use linear regression, then the compatibility definition itself does not have much advantage; I would consider this more useful if we had results over several algorithms and we understand which is good in what setting. Also, due to the simplicity of linear regression with gradient descent, it looks like the compatibility region tells us more about when to stop for how kind of data generating distribution, which is an overstudied topic.
- The literature summary is not very thorough in my view, especially compared to the rest of the paper. PAC-Bayes generalization theory is almost nowhere discussed, however, there are so many new studies that may be doing what this paper is doing implicitly. Except for an analysis in the Appendix, I did not see much on Rademacher complexity-based generalizations, and I would like to know more about how these approaches differ.
- The numerical experiments are weak. There is not much these experiments tell the reader, rather than displaying mostly what was shown. The selection of parameters is also done in an ad-hoc manner.
- I am not sure if considering the data-algorithm pair was *completely* not studied. This is a very strong and risky statement. I agree that in generalization theory this is studied much less, though.

***Minor Weaknesses/Questions***
- Can we get have any finite sample guarantees? This is very popular these days. I guess it should be easier to analyze in the overparameterized setting.
- Abstract: The Nagarajan citation is on Deep Learning, but the sentence sounds general.
- Abstract: "more suitable notion" -> more than what?
- Abstract, and anywhere else: "last iterate analysis" is either not cited or not explained. This looks vague.
- Introduction has a sentence that says "both the training algorithm and the data distribution play essential roles in generalization analysis". This is well-known for any setting, not just overparameterized regime.
- Introduction and overall: The question "How to incorporate both data factor and algorithm into generalization analysis?" is not motivated. Why should we be interested in this? For example, later on, you show that this would require fewer assumptions. This might be one reason. But overall, please answer "why" in addition to "how".
- The introduction has "the final interpolator ..." this term and sentence stay unclear.
- Figure 1 (a): the right axis is different than the left axis. Why is that?
- Page 2: The sentence that starts with "Informally speaking," is not clear to me.
- Section 3.1, **Algorithm**: The parameter $t$ has almost no explanation. This is not clear at all. In general, we keep referring to $t$ in Definition 3.1, too, but this only makes sense in the gradient descent setting. The notation or motivation is not clear.
- Definition 3.1: As stated above, $\boldsymbol{\theta}_{n}^{(t)}$ is not clear. Could you also somewhere state why such an asymptotic convergence in probability would imply generalization? $T_n$ notation is not clear until someone comes to the next sections.
- Could you please relate **Assumption 3** to the fact that $(\boldsymbol{X}\boldsymbol{X}^\top)$ is invertable?
- "iff" -> if and only if
- Page 6 refers to Theorem 5.1, this should be a typo.
- Overall too many times Vartless et al. (2019) are cited. Could you please mention it less? Maybe we can mention benign overfitting first, and then keep referring to "benign overfitting" instead?
- In the Conclusion and throughout the paper the authors say "this method eases the assumptions". However, I am not sure about one thing. Here, since we consider algorithm and data together, we are somehow splitting the assumptions equally over them. If we had not parametrized one of either, then we would have needed more assumptions on the latter. In other words, having control over both already gives us the power to decrease the severity of assumptions.
- The **Setup** part in Section 6 is very simplistic. In general, it is not "fun" to read these experiments. They are very standard. I would appreciate seeing more original experiments. The selection of parameters is very ad-hoc and not clear.

**Summary Of The Paper:**

This work has a key focus on generalization in overparameterized learning problems. The authors propose a new trajectory analysis for generalization that depends both on the data-generating distribution and the algorithm of the underlying setting, unlike most of the existing work which typically concentrates on one of either. A condition, named *compatibility* is being introduced, which asserts that a data-generating distribution and an algorithm are compatible roughly when the excess risk converges (both data and algorithm are treated random, as a key component of this work)  to zero as the training sample size grows. A *compatibility region* is the collection of "allowed" values that the data generating distribution and the algorithm can be parametrized with where the resulting pair is compatible. The authors' main focus after this definition is to derive the compatibility region of a linear regression model that is trained with gradient descent (number of iterations as a parameter that belongs in the compatibility region) with the underlying feature-label pairs. Some of the claims are exhibited via synthetic numerical experiments, and then comparisons with some of the existing bounds are analyzed.

**Summary Of The Review:**

The literature review, writing of the paper, numerical experiments, and motivation are poor. However, the paper has a thorough analysis, strong results, and looks mathematically correct.

---

> ### Author Response · Authors · 2022-11-09
> **Clarifications (1)**
>
> We thank the reviewer for the valuable, insightful and detailed review. The provided suggestions are extremely helpful and constructive. We revise the paper accordingly, which is marked by **blue**. Below, we do our best to address the reviewer's questions and respond to the reviewer's detailed remarks.
>
> >Q1: The paper after Definition 3.1 is more or less all about linear regression with gradient descent. [...] I am therefore wondering, to what extent can we use it for other models. [...] Also, due to the simplicity of linear regression with gradient descent, it looks like the compatibility region tells us more about when to stop for how kind of data generating distribution, which is an overstudied topic.
>
> A1: We thank the reviewer for the valuable comments. We consider overparameterized linear regression in this paper for understanding the compatibility. However, the notion compatibility is much more general and can be non-linear. Studying overparameterized linear regression is helpful in understanding the complex scheme in deep learning since (a) it exhibits phenomena that are also observed for more complex models, e.g., the benefit of early stopping (see Figure 1 in our paper, and [1,2]), and (b) it is correlated to neural networks under the Neural Tangent Kernel framework. We also extend the results to  kernel regime in Appendix B.6. We finally remark that deriving a large compatibility region in such a regime is non-trivial, since it requires a delicate analysis on the training trajectory.
>
>
> >Q2: The literature summary is not very thorough in my view [...] PAC-Bayes generalization theory is almost nowhere discussed [...] Except for an analysis in the Appendix, I did not see much on Rademacher complexity-based generalizations[...]
>
> A2: We thank the reviewer for the suggestions. As one of the most exciting directions in generalization analysis, PAC-Bayes theory performs well empirically and theoretically, and can even yield non-vacuous bounds in deep learning regimes. However, PAC-Bayes theory cannot be applied in our regime since it analyzes a distribution over classifiers (prior/posterior distribution). In contrast, our results focus more on the classifiers on the training trajectory, which is not necessarily a distribution over classifiers. We add the discussion of PAC-Bayes theory in the related work part.
>
> We next consider the Rademacher complexity-based techniques, which are among the most commonly used methods in generalization community. Compared to them, The notion compatibility extract more algorithm information by focusing more on the trajectory of the training algorithm. Therefore, we have a better compatibility region than theirs (see appendix B.4)
>
> >Q3: The numerical experiments are weak. There is not much these experiments tell the reader, rather than displaying mostly what was shown. The selection of parameters is also done in an ad-hoc manner.
>
> A3: We thank the reviewer for the valuable comment. Indeed, the experiment section is to motivate and validate our theoretical results. In the revision, we add experiments with various hyperparameters (for example, the data dimension, sample size, data covariance) for the linear regression experiments. See appendix C.1. for details. We hope this can relieve the reviewer's concern. We are eager to conduct any further experiments if the reviewer has any further questions.
>
> >Q4: I am not sure if considering the data-algorithm pair was completely not studied. This is a very strong and risky statement. I agree that in generalization theory this is studied much less, though.
>
> A4: This is a very interesting question. We refer to the second paragraph in the general response for more details.

---

> > ### Author Response · Authors · 2022-11-09
> > **Clarifications (2)**
> >
> > >Q5: Can we get have any finite sample guarantees? This is very popular these days. I guess it should be easier to analyze in the overparameterized setting.
> >
> > A5: Although the proposed notion of compatibility requires $n \to \infty$, Theorem 5.1 does not require it and has therefore a finite sample guarantee.
> >
> > >Q6: Abstract: The Nagarajan citation is on Deep Learning, but the sentence sounds general.
> >
> > A6: Thanks for the suggestion. We have revised the sentence into: *One of the major open problems in machine learning is to characterize generalization in the overparameterized regime, where most traditional generalization bounds become inconsistent *even for overparameterized linear regression* (Nagarajan and Kolter, 2019).*
> >
> > >Q7: Abstract: "more suitable notion" -> more than what?
> >
> > A7: Here the comparison is between compatibility and last-iterate analysis. We have revised the sentence to remove the confusion: *By considering the entire training trajectory and focusing on early-stopping iterates, compatibility exploits the data and the algorithm information and is therefore a suitable notion for generalization of overparameterized models.*
> >
> > >Q8: Abstract, and anywhere else: "last iterate analysis" is either not cited or not explained. This looks vague.
> >
> > A8: We have added the citation in the second paragraph in the introduction section. In overparameterized linear regression setting, last iterate analysis is equivalent to min-norm solution analysis, considered in [3].
> >
> > >Q9: Introduction has a sentence that says "both the training algorithm and the data distribution play essential roles in generalization analysis". This is well-known for any setting, not just overparameterized regime.
> >
> > A9: Thanks for the advice, and we have revised the sentence into: *In generalization analysis, both the training algorithm and the data distribution play essential roles.*
> >
> >
> > >Q10: Introduction and overall: The question "How to incorporate both data factor and algorithm into generalization analysis?" is not motivated. Why should we be interested in this? For example, later on, you show that this would require fewer assumptions. This might be one reason. But overall, please answer "why" in addition to "how".
> >
> > A10: We have added the following sentence before the question: *Combining both data and algorithm factor into generalization analysis can help deriving tighter generalization bound and explaining the generalization ability of overparameterized models observed in practice.* We wish this could strength our motivation.
> >
> > >Q11: The introduction has "the final interpolator ..." this term and sentence stay unclear.
> >
> > A11: We change the term the final interpolator to the final iterate. We have revised the sentence into: In the MNIST case, the final iterate on the SGD trajectory has 19.9\% test error, much higher than the 2.88\% test error of the best iterate on the trajectory.
> >
> > >Q12: Figure 1 (a): the right axis is different than the left axis. Why is that?
> >
> > A12: If we use the same scale in the left and right axis in figure 1a, the trend of the training loss will become hard to read.
> >
> > >Q13: Page 2: The sentence that starts with "Informally speaking," is not clear to me.
> >
> > A13: We have revised the sentence into: *Informally speaking, an algorithm is compatible with a data distribution if as the sample size goes to infinity, the minimum excess risk of the iterates on the training trajectory converges to zero.* We hope that this will clarify the concerns.
> >
> > >Q14: Section 3.1, Algorithm: The parameter t has almost no explanation. This is not clear at all. In general, we keep referring to t in Definition 3.1, too, but this only makes sense in the gradient descent setting. The notation or motivation is not clear.
> >
> > A14: We apologize for the confusion. The parameter $t$ represents the iteration number of the iterative training algorithm. We have revised the paragraph accordingly.
> >
> > >Q15: Definition 3.1: As stated above, $\theta_n^{(t)}$ is not clear. Could you also somewhere state why such an asymptotic convergence in probability would imply generalization? $T_n$ notation is not clear until someone comes to the next sections.
> >
> > A15: The convergence in probability argument implies that for any failure tolerance $\varepsilon$, the excess risk on $T_n$ is small with high probability $1-\varepsilon$, as long as the sample size $n$ is larger than a certain threshold. Therefore, if we can early stop in the region $T_n$, the model will generalize well.
> >
> > We have also clarified definition of $\theta_n^{(t)}$ and $T_n$ in Definition 3.1 in the revision.

---

> > > ### Author Response · Authors · 2022-11-09
> > > **Clarifications (3)**
> > >
> > > >Q16: Could you please relate Assumption 3 to the fact that $XX^\top$ is invertable?
> > >
> > > A16：The two assumptions are equivalent.
> > > Suppose $XX^\top$ is not invertible, then we know that there exists $u\in \mathbb{R}^{n}$, such that $u^\top XX^\top u=0$. This indicates $||Xu||=0$, $Xu=\mathbf{0}$, $\sum_{i=1}^n u_i x_i=\mathbf{0}$, contradicting the linear independency assumption in Assumption 3. Conversely, if the $x_1,\cdots, x_n$ are not linearly independent, there exists  $u\in \mathbb{R}^{n}$, such that $\sum_{i=1}^n u_i x_i=\mathbf{0}$, $Xu=\mathbf{0}$, $u^\top XX^\top u=0$, contradicting the inveribility of $XX^\top$.
> > >
> > > We have made the corresponding adjustments in the paragraph before Assumption 3.
> > >
> > > >Q17: "iff" -> if and only if
> > >
> > > A17: We have made the revision.
> > >
> > > >Q18: Page 6 refers to Theorem 5.1, this should be a typo.
> > >
> > > A18: Thanks for kindly pointing this out. We have corrected the typo.
> > >
> > > >Q19: Overall too many times Vartless et al. (2019) are cited. Could you please mention it less? Maybe we can mention benign overfitting first, and then keep referring to "benign overfitting" instead?
> > >
> > > A19: Thanks for the suggestion. We have removed some of the citations.
> > >
> > > >Q20: In the Conclusion and throughout the paper the authors say "this method eases the assumptions". However, I am not sure about one thing. Here, since we consider algorithm and data together, we are somehow splitting the assumptions equally over them. If we had not parametrized one of either, then we would have needed more assumptions on the latter. In other words, having control over both already gives us the power to decrease the severity of assumptions.
> > >
> > > A20: We thank the reviewer for the insightful question!
> > > The compatibility-based analysis ease the assumption in the sense that: if the previous generalization analysis holds, the compatibility region cannot be empty. Existing generalization tasks mainly focus on a given fixed time, e.g., the last iterate, while the compatibility notion focuses on finding a non-empty time region. Therefore, if for a problem instance there exists a iterate that can generalize well, it must have a nonempty compatibility region. In this process, the algorithm information is encoded in deriving such a compatibility region.
> > >
> > >
> > > >Q21: The Setup part in Section 6 is very simplistic. In general, it is not "fun" to read these experiments. They are very standard. I would appreciate seeing more original experiments. The selection of parameters is very ad-hoc and not clear.
> > >
> > > A21: We have moved the setups to the appendix. Thanks for the suggestion.
> > >
> > >
> > >
> > > Finally, we would like to thank the reviewer once again for the effort in providing us valuable and helpful suggestions. We will, of course, remain at the disposal if the reviewer has any further questions, and continue to provide clarifications as promptly as possible.
> > >
> > >
> > >
> > >
> > >
> > >
> > >
> > > [1] M. Li, M. Soltanolkotabi, and S. Oymak. Gradient descent with early stopping is provably robust to label noise for overparameterized neural networks.
> > >
> > > [2] Z. Ji, J. D. Li, and M. Telgarsky. Early-stopped neural networks are consistent.
> > >
> > > [3]P. L. Bartlett, P. M. Long, G. Lugosi, and A. Tsigler. Benign overfitting in linear regression.

---

> > > > ### Comment · Reviewer_a4RZ · 2022-11-17
> > > > **Acknowledging the authors' response**
> > > >
> > > > Dear Authors,
> > > >
> > > > I would like to thank you for this excellent review. Your responses are extremely clear, and the revision includes almost all of the points discussed. I also read your responses to other reviewers, and I do not have further questions.
> > > >
> > > > Some of my previous questions (e.g., Q16 above) were not direct questions to you, but rather kind requests for clarification, but I can see you did both: answered it here **and** updated the document. To this end, I believe the new structure of the paper is great and easier to read.
> > > >
> > > > I remain positive about this paper, and I believe this is good work.
> > > >
> > > > I still believe the paper provides *an* alternative generalization perspective and compared to the existing ones this is still in the early stage, and I am not 100% confident how the researchers in this field will feel about the method proposed here. The discussion on the PAC-Bayes comparison is still not very convincing to me -- the updated paper does not have a sufficient discussion, and I don't understand what additional benefit comes from "trajectory analysis over distribution over classifiers".
> > > >
> > > > However, given the thorough analyses and great review, I would like to give the authors the benefit of the doubt and I will keep my positive score, whilst carefully following discussions with the other reviewers.
> > > >
> > > > Thank you for your time!

---

> > > > > ### Author Response · Authors · 2022-11-17
> > > > > **Thank You!**
> > > > >
> > > > > We sincerely appraciate the reviewer's detailed, constructive and positive reply. Below we provide further comparisons with PAC-Bayes methods. We hope that these clarifications could further elimate the reviewer's concerns. We have revised our manuscript with the following discussion in Appendix B.11.
> > > > >
> > > > > As one of the most exciting techniques in generalization analysis, PAC-Bayes theory works both theoretically and empirically. Usually, the form of PAC-Bayes bounds relies on the distance between the *prior distribution* which is unrelated to the training procedure, and the  *posterior distribution* which is the distribution after training (*e.g.*, isotropic Gaussian distribution centered at a trained parameter).
> > > > >
> > > > > In this sense, PAC-Bayes considers different regimes from this paper, since the returned classifier in this paper is not a distribution.
> > > > > Moreover, PAC-Bayes usually do not explicitly focus on early-stopping during analysis, while the notion of compatibility heavily relies on the early-stopping arguments.
> > > > > By considering early-stopping that encodes algorithm information, we can derive tighter generalization bound relying on weaker assumptions.
> > > > >
> > > > > *More interestingly, PAC-Bayes framework is not mutually exclusive with the trajectory analysis.*
> > > > > One can indeed introduce trajectory analysis into PAC-Bayes techniques and derive a compatibility region based on trajectory-based PAC-Bayes theory.
> > > > > By doing so, one can explicitly incorporate more algorithm information into PAC-Bayes framework.
> > > > > We leave more detailed discussion for future work.
> > > > >
> > > > > We next show the differences in more detail:
> > > > >
> > > > > - **Different settings.** In PAC-Bayes analysis, the returned classifier is a distribution instead of a fixed parameter. Forcing PAC-Bayes analysis in the fixed parameter regime would cost much because it is hard to define distribution distance (*e.g.*, KL divergence) on two *single-point distributions*.
> > > > >
> > > > > - **Different classifiers.** Although both methods consider 'a bag of' classifiers, they are fundamentally different. In PAC-Bayes framework, the trained random classifiers are regarded to be *independently* drawn from the posterior distribution. However, in trajectory analysis, all classifiers are *dependent* during the training process. Therefore, trajectory analysis is more challenging in this sense.
> > > > >
> > > > > - **Different characterizations.** PAC-Bayes framework characterizes the *expectation* of generalization loss, over the randomness of the posterior distribution over parameters, which do not explicitly sketch the time factor. In comparison, the trajectory analysis in this paper focus on providing a *compatibility region*, where the generalization error is uniformly consistent and considers the time factor.
> > > > >
> > > > > We would like to express our gratitude again to the reviewer for the constructive suggestions and positive feedback. We are eager to provide any further clarifications to help the evaluation. If we have successfully addressed the concerns, we would sincerely appreciate an increased score, if possible.

---

> > > > > > ### Comment · Reviewer_a4RZ · 2022-12-05
> > > > > > **Thanks for the PAC-Bayes Clarification**
> > > > > >
> > > > > > Dear Authors,
> > > > > >
> > > > > > Thank you very much for your responses and apologies for my late reply. I believed after the discussion period my responses will not be posted, but I can see the reply button still works!
> > > > > >
> > > > > > In general, when I first read the paper, such comparisons with the literature were not immediately clear and I found it hard to believe that the setting of this paper is "never addressed" in the literature. It is great to read about the clarification at least on the PAC-Bayes theory.
> > > > > >
> > > > > > I still like the analysis of this paper and keeping my marginally positive score.
> > > > > >
> > > > > > Best regards.

---

> > > > > > > ### Author Response · Authors · 2022-12-07
> > > > > > > **Thank You!**
> > > > > > >
> > > > > > > We thank the reviewer for the active discussions which help us enhance our manuscript considerably. Again, we would like to express our greatest gratitude for the reviewer's detailed and constructive comments. Thanks!

---

### Author Response · Authors · 2022-11-09
**General Response**

We appreciate the reviewers for their constructive comments that help to enhance our manuscript considerably. Furthermore, we would like to express our gratitude to the reviewers for their positive feedback on this paper's novelty, which has been a great source of inspiration for us.

Some reviewers are concerned that previous techniques also consider both data and algorithm information. This is indeed true. However, in this paper, the taxonomy of data-dependent / algorithm-dependent technique does not mean that they totally ignore all the information of algorithm/data, but mean that it loses some important algorithm/data information which makes it vacuous/inconsistent in generalization analysis. For example, uniform convergence indeed uses some kind of algorithm information (e.g., parameter norm $|| \theta ||$) but it is still classfied as a data-dependent technique under this criterion [1]. We add a discussion in Footnote 1 and revise our statements accordingly.

Besides, the most critical comment comes from Reviewer rKAH, who mentioned that the results can be covered by the existing literature (Yao et al.[2]). However, it is not true. Their results gives a fast rate (which is minimax at the optimal stopping time) while our results give a larger compatibility region (the time interval in which the excess risk is consistent) . In our regimes, compatibility region is meaningful since it sketches how easy a trained model can be consistent. Speficially, we can obtain a compatibility region like $t \in (0, n)$ while their results can only lead to a compatibility region like $t \in (0, \sqrt{n})$. Threrefore, our results cannot be covered. We have added a discussion in Appendix B.5 in the revision.

[1] Zhang, C., Bengio, S., Hardt, M., Recht, B., and Vinyals, O. (2021). Understanding deep learning (still) requires rethinking generalization.

[2] Yao, Y., Rosasco, L., and Caponnetto, A. (2007). On early stopping in gradient descent learning.

---

### Decision · Program_Chairs · 2023-01-20

**Decision:**

Reject

**Justification For Why Not Higher Score:**

Based on all the information provided above, together with the reviewers, I came to conclusion that in its current for the paper can not be accepted. Even though the paper provides a significant and interesting contribution (to the field of benign overfitting in linear regression), it motivates and frames its results in a confusing and misleading way. A major revision is required before the paper can be considered for acceptance.

**Justification For Why Not Lower Score:**

N/A

**Metareview: Summary, Strengths And Weaknesses:**

The paper's main contribution, claimed by the authors, is the introduction of "data-algorithm compatibility" (Definition 3.1). The iterative learning algorithm and a data distribution are called compatible whenever the excess risk of the early-stopped version of the algorithm converges to zero with high probability. This concept is meant to provide a much better account for the properties of the algorithm, compared to the existing literature. As the first step towards establishing the usefulness and significance of the compatibility notion, the authors provide results for the linear over-parametrized regression with early stopping (Theorem 4.1), that improves previous results on benign overfitting (Bartlett et. al. 2019). Previous literature on benign overfitting focuses on the last iterate (min-norm) solution. By also considering all the intermediate steps, the authors manage to relax the assumptions required for the convergence: in particular, extend it to the data with polynomially decaying eigenvalues.

The reviewers identified Theorem 4.1 as the main contribution of the paper. They all agreed that it provides a concrete and interesting contribution to the field of benign overfitting in overparametrized linear regression by accounting for the early stopping, which was not done before.

Unfortunately, after the detailed rebuttal phase and a virtual meeting, all the reviewers also unanimously agreed that there is an issue with the way the paper presents and motivates its contributions. The whole story and discussion in the paper is built around significance and novelty of the "data-algorithm compatibility", while the application to benign overfitting (Theorem 4.1) is presented as merely a first step in establishing this claimed significance. The reviewers noticed that "centering the paper around the notion of *compatibility* as a way of encoding algorithm and data-dependence is not as valuable and significant in itself as the paper claims". Most of the reviewers agreed that Definition 3.1 goes one step beyond in capturing the properties of the algorithm, compared to the existing results. However, the only additional aspect captured by Definition 3.1 is the early stopping. It indeed captures more than many of the existing bounds, but, similarly, one could come up with scenarios where the "important algorithm information" is not captured by the early stopping. Given the only application of the compatibility, presented by the authors, is in the context of linear over-parametrized models with early stopping, and given the proof techniques look too tailored to the analysis of linear models, the reviewers were sceptical about general applicability and significance of Definition 3.1. The reviewers suggested "to rethink the story of the paper and focus on highlighting the key technical contributions [Theorem 4.1, early stopped overparametrized linear regression and benign overfitting] rather than the abstraction of compatibility --- unless, there are multiple applications of compatibility beyond linear regression that can somehow drive home the point that this abstraction is insightful".

Considering all these points, I have to recommend rejection.

**Summary Of Ac-Reviewer Meeting:**

We went into the video conference meeting with three borderline accepts and one borderline reject. Given the score was borderline and no one stepped up to champion the paper, I wanted to make sure there was nothing we were missing.

First of all, the reviewer voting for "borderline reject" identified, that they would not fight for the reject. In their original review, they thought that the results presented by the authors were already known, but that confusion was cleared after the rebuttal.

We all established that Theorem 4.1 did indeed provide a novel contribution to the particular field of the linear overparametrized regression and benign overfitting by accounting for early stopping. We all think that it is a significant and solid contribution.

Then we spent time discussing the main claimed contribution of the paper --- the concept of compatibility. All the reviewers shared that the abstract and introduction made them excited about the proposed concept of compatibility: they were expecting something approach that would allow to account for both properties (data / algorithm). They all felt the spotlight was on the compatibility notion.

Finally, we spent a good amount of time discussing if the evidence and results provided by the authors were convincing enough to suggest that the notion of compatibility deserved a central role in the paper. It was concluded that, unfortunately, not.